# FRISM: Fine-Grained Reasoning Injection via Subspace-Level Model Merging for Vision–Language Models

Chenyu Huang [* 1]  Peng Ye [* 2 3]  Xudong Tan [1]  Jinhan Mu [1]  Shenghe Zheng [4]  Li Shen [5]  Tao Chen [1 6]

## Abstract

Efficiently enhancing the reasoning capabilities of Vision-Language Models (VLMs) by merging them with Large Reasoning Models (LRMs) has emerged as a promising direction. However, existing methods typically operate at a coarse-grained layer level, which often leads to a trade-off between injecting reasoning capabilities and preserving visual capabilities. To address this limitation, we propose FRISM (Fine-grained Reasoning Injection via Subspace-level model Merging), a fine-grained reasoning injection framework based on subspace-level model merging. Observing that different SVD subspaces contribute differently to reasoning and perception, FRISM decomposes LRM task vectors via Singular Value Decomposition (SVD) and adaptively tunes the scaling coefficients of each subspace through learning to realize fine-grained reasoning injection. Furthermore, we introduce a label-free self-distillation learning strategy with dual-objective optimization using common vision-language perception datasets. Extensive experiments demonstrate that FRISM effectively improves reasoning capabilities while largely preserving the model's visual capabilities by consistently achieving strong performance across diverse visual-language reasoning benchmarks.

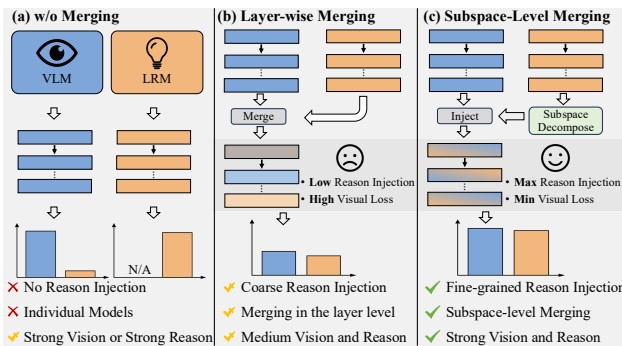

*Figure 1.* Illustration of different model merging strategies.

## 1. Introduction

In recent years, represented by LLaVA (Liu et al., 2023; 2024a), Qwen-VL (Wang et al., 2024b; Bai et al., 2025), and InternVL (Chen et al., 2024b; Zhu et al., 2025), Vision Language Models (VLMs) built upon Large Language Models (LLMs) have achieved remarkable success, demonstrating strong performance across a wide range of vision language tasks, including image recognition, visual understanding, and text generation (Liu et al., 2023; 2024a; Cao et al., 2024; Brooks et al., 2023). In parallel, the emergence of large language reasoning models (LRMs) like Deepseek-R1 (Guo et al., 2025) and OpenAI-o1 (Jaech et al., 2024) has significantly advanced the frontier of model capabilities, especially in complex tasks like mathematical reasoning, problem solving, programming, and verification (Chen et al., 2025b; Zheng et al., 2026). Accordingly, many recent studies have focused on improving the reasoning capabilities of VLMs (Wang et al., 2025c), proposing various benchmarks (Yang et al., 2025a; Wang et al., 2024a) and training strategies (Wang et al., 2025a; Huang et al., 2025). These works consistently show the effectiveness of enhancing reasoning capabilities to improving the performance of VLMs.

Due to the lack of labeled vision-language reasoning training datasets and the heavy post-training costs, efficiently equipping VLMs with reasoning capabilities has become an important research direction. Among existing methods (Xiao et al., 2026; Guan et al., 2026; Huang et al., 2025), model merging achieves this by integrating the weights of LRMs and VLMs, incurring minimal training overhead and data annotation costs. Bring Reason to Vision (BR2V) (Chen et al., 2025c) first demonstrates the feasibility of reasoning injection from LRMs to VLMs through model

*Equal contribution [1]College of Future Information Technology, Fudan University, Shanghai, China [2]Shanghai Artificial Intelligence Laboratory, China [3]The Chinese University of Hong Kong, China [4]Harbin Institute of Technology, China [5]Sun Yat-Sen University, Shenzhen, China [6]Shanghai Innovation Institute, China. Correspondence to: Tao Chen <eetchen@fudan.edu.cn>.

*Proceedings of the 43rd International Conference on Machine Learning*, Seoul, South Korea. PMLR 306, 2026. Copyright 2026 by the author(s).

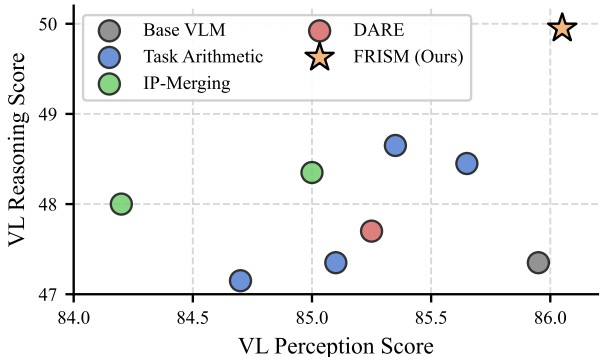

*Figure 2.* Vision-Reasoning tradeoff when merging VLMs and LRMs. Task Arithmetic and IP-Merging are applied under different merging coefficients and similarity thresholds, respectively.

merging. However, common model merging techniques are developed for multi-task settings, which may compromise the visual capabilities of the merged models and cause them to fail. To this end, FRANK (Wei & Chen, 2025) proposes a layer-wise model merging method by identifying shallow and deeper layers and employing a Taylor-derived closed-form solution to guide the merging process. IP-Merging (Hu et al., 2025) only merges a subset of typical layers based on the similarity between the weights of VLMs and LRMs while leaving the remaining layers unchanged. Although effective to some extent, these methods still exhibit a significant performance gap compared to post-training-based reasoning VLMs.

To minimize this gap, we rethink the merging of LRMs and VLMs from the perspective of preserving both reasoning and visual capabilities. Existing merging methods mainly operate at the layer level and can be formulated as:

$$W_{\mathrm{M}} = W_{\mathrm{base}} + \sum_{m \in \{\mathrm{vlm,lrm}\}} \lambda_m \cdot \tau_m, \qquad (1)$$

where $W_{\mathrm{M}}$ represents the weights of the merged layer, and $W_{\mathrm{base}}$ denotes the weights of the pre-trained base model. The term $\tau_m$ refers to the task vector derived from model $m$, and $\lambda_m$ is the associated merging coefficient, where $m \in \{\mathrm{vlm,lrm}\}$ corresponds to VLM and LRM, respectively. Fig. 1 further shows the diagram of (a) individual models, (b) layer-wise merging. For individual models, $\lambda_{\mathrm{vlm}} = 1, \lambda_{\mathrm{lrm}} = 0$ recovers the original VLM, while $\lambda_{\mathrm{vlm}} = 0, \lambda_{\mathrm{lrm}} = 1$ recovers the original LRM. Layer-wise merging methods assign layer-dependent values to $\lambda_{\mathrm{vlm}}$ and $\lambda_{\mathrm{lrm}}$, where a larger $\lambda_{\mathrm{vlm}}$ indicates stronger visual capability and a larger $\lambda_{\mathrm{lrm}}$ indicates stronger reasoning capability. However, such layer-wise balancing is inherently coarse-grained and can only produce a compromise: tuning $\lambda_{\mathrm{vlm}}$ and $\lambda_{\mathrm{lrm}}$ at the layer level entangles the two capabilities, leading to unavoidable losses in either vision or reasoning

capabilities. Fig. 2 shows the average performance on two vision-language (VL) reasoning datasets, MMStar (Chen et al., 2024a) and R1-OneVision (Yang et al., 2025a), and two VL perception datasets, TextVQA (Singh et al., 2019) and POPE (Li et al., 2023b). The results show that existing methods usually lead to a trade-off between VL reasoning and perception performance.

In this paper, we observe that a layer is not the atomic unit of model capability: even within a single layer, reasoning capabilities and visual representations can reside in distinct subspaces. Motivated by the low-rank nature and the compact encoding of high-level or reasoning-related functions of parameter updates in LLMs (Cai et al., 2025; Ping et al., 2024; Zhou et al., 2025), we use the subspaces of the LRM task vectors obtained through singular value decomposition (SVD) as the reasoning prior and tune the injection magnitude of each subspace. We find that the gains from different ranks peak at different magnitudes. This heterogeneity suggests that a uniform layer-wise scaling coefficient is insufficient: it inevitably couples beneficial reasoning, which may require high magnitude, with visual noise, thereby limiting overall improvements.

Thus, to achieve the maximum reasoning capabilities with the minimum degree of vision loss, we propose FRISM (Fine-grained Reasoning Injection via Subspace-level model Merging). As shown in Fig. 1 (c), we change the merging paradigm to:

$$W_{\mathrm{M}} = W_{\mathrm{base}} + \tau_{\mathrm{vlm}} + \sum_i \lambda_i \cdot B_i, \quad B_i \subseteq \tau_{\mathrm{lrm}} \quad (2)$$

where $B_i^l$ refers to the rank-$i$ subspace of reasoning task vectors. Specifically, we fix the subspaces from the LRM task vectors and tune the scaling coefficient $\lambda_i^l$ for each subspace. To determine optimal coefficients without relying on scarce multimodal reasoning annotations, we introduce a label-free multimodal self-distillation mechanism. A compound objective is introduced to balance two goals: minimizing the KL divergence to preserve visual perception capabilities and maximizing the spectral magnitude of the injected subspaces to encourage the effective assimilation of reasoning priors. Thus, the model learns to selectively absorb reasoning capabilities through prior subspaces while preserving the original visual performance. The comparison of the performance between our FRISM and existing methods is shown in Fig. 2, demonstrating that FRISM improves reasoning performance while preserving visual capabilities. Our contributions can be summarized as follows:

- We rethink the paradigm of merging VLMs and LRMs to efficiently enhance visual reasoning capabilities. Instead of coarse-grained merging at the layer-level, we propose a fine-grained merging paradigm to realize more effective reasoning injection.

- We first propose a subspace-level model merging for reasoning injection from LRMs to VLMs. By decomposing LRM task vectors via SVD and modulating at the subspace granularity, our method enables the adaptive reweighting of SVD subspaces under a visual-preservation objective.

- We design label-free self-distillation training to relieve the scarcity of multimodal reasoning data. Through a dual-objective optimization that safeguards visual consistency while maximizing spectral injection intensity, our approach achieves stable and efficient reasoning transfer without requiring VL reasoning datasets.

- Our method serves as a versatile, plug-and-play solution compatible with various VLM architectures and scales. Extensive experiments demonstrate that it consistently achieves state-of-the-art performance across diverse models and reasoning benchmarks, validating its robustness and effectiveness.

## 2. Related Work

**Model Merging** integrates models finetuned on different tasks within the parameter space to assimilate diverse capabilities (Yang et al., 2026). In multi-task settings, the critical challenge for model merging is weight interference. Various methods are proposed to address this issue, including calculating information matrices (Jin et al., 2023; Matena & Raffel, 2022), merging coefficient tuning (Ilharco et al., 2023; Yang et al., 2024c), task vector pruning (Yadav et al., 2023; Yu et al., 2024; Huang et al., 2024; Gargiulo et al., 2025; Marczak et al., 2025), and dynamic routing (Ye et al., 2025; Tang et al., 2024; Lu et al., 2024b; Shen et al., 2026; Yang et al., 2024b), Recently, this paradigm has been extended to merge VLMs and LRMs (Chen et al., 2025c), aiming to transfer the reasoning capabilities between models trained on different modalities. However, methods designed under multi-task settings often incur severe performance loss. To relieve this issue, FRANK (Wei & Chen, 2025) performs layer-wise weighted fusion based on Taylor expansion and IP-Merging (Hu et al., 2025) mitigates cross-modal conflicts via parameter projection based on weight similarity. Nevertheless, limited by coarse granularity or static thresholding, these methods struggle to distinguish parameters beneficial for reasoning from those that cause visual interference. Therefore, effective Reasoning-to-Vision transfer requires not merely conflict mitigation but also precise control over task vectors. To this end, we propose FRISM, which finegrainedly localizes and preserves parameter components that are critical for reasoning without harming visual capabilities.

**Subspace Decomposition** technique aims to factorize high-dimensional matrices to reveal latent structures. Methods such as Non-negative Matrix Factorization (NMF) and Independent Component Analysis (ICA) demonstrate advantages in interpretability or disentanglement. Among them, SVD serves as a fundamental subspace decomposition technique offering optimal low-rank approximations. It stands as a cornerstone in deep learning for model compression (Idelbayev & Carreira-Perpiñán, 2020) and parameter-efficient fine-tuning (PEFT) (Hu et al., 2022; Meng et al., 2024). Recent studies (Cai et al., 2025; Sharma et al., 2024; Zhang et al., 2023; Ping et al., 2024; Yang et al., 2025b) indicate that model capabilities are not uniformly distributed but are distinctively encoded within specific rank subspaces. Motivated by these findings, we treat LRM task vectors as inference priors. By recalibrating their spectral distribution via SVD, FRISM seamlessly adapts textual reasoning priors to the VL domain, thereby enhancing VLM reasoning capabilities without compromising visual performance.

## 3. Method

**Preliminaries.** Consider a VLM $\theta_{vlm}$ and an LRM $\theta_{lrm}$ finetuned from a pre-trained base model $\theta_{base}$. Task vectors are defined to be the difference between the post-trained models and the pre-trained models, thus the task vectors for VLMs and LRMs can be respectively computed by:

$$\tau_{vlm} = \theta_{vlm} - \theta_{base}; \quad \tau_{lrm} = \theta_{lrm} - \theta_{base}, \quad (3)$$

where $\tau$ are task vectors. Existing methods usually conduct model-level merging (Chen et al., 2025c):

$$\theta_{vlrm} = \theta_{base} + \lambda_{vlm} \cdot \tau_{vlm} + \lambda_{lrm} \cdot \tau_{lrm}, \quad (4)$$

or layer-level merging (Hu et al., 2025; Wei & Chen, 2025):

$$\theta_{vlrm}^{(l)} = \theta_{base}^{(l)} + \lambda_{vlm}^{(l)} \cdot \tau_{vlm}^{(l)} + \lambda_{lrm}^{(l)} \cdot \tau_{lrm}^{(l)}, \quad l = 1..L \quad (5)$$

for models with $L$ layers. Note that during the merging procedure, only the LLM part of VLMs is applied for merging, while the vision-related layers (e.g., visual tower and visual projection layers) remain fixed.

**Motivation.** We hypothesize that reasoning capabilities and visual representations reside in distinct subspaces within the parameter space. Consequently, a coarse-grained, layer-wise merging coefficient may be insufficient to disentangle these capabilities. To validate this hypothesis, we analyze the impact of merging different subspaces of the LRM task vector. Specifically, we decompose the task vector of DeepSeek-R1-Distill-Qwen-7B, via SVD and inject specific rank subspaces into the VLM, Qwen2.5-VL-Instruct, with varying scaling coefficients. We observe that (1) different subspaces achieve peak performance at distinctly different scaling coefficients; (2) standard layer-wise merging results in suboptimal performance. These observations suggest that

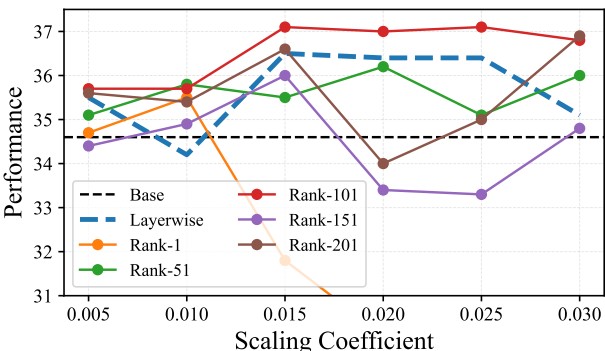

*Figure 3.* Impact of scaling coefficients across subspace ranks when merging Qwen2.5-VL-7B-Instruct and the subspaces of DeepSeek-R1-Distill-Qwen-7B task vector.

subspace-level modulation is a useful granularity for reasoning injection. Therefore, to achieve maximum reasoning capabilities with minimal vision loss, it is essential to shift from coarse-grained layer-level merging to a fine-grained subspace-level merging paradigm. The detailed configuration of Fig. 3 is shown in Appendix B. Additional results demonstrating our findings can be found in Appendix D.3.

### 3.1. FRISM: Adaptive Subspace-level Merging

The framework of the proposed FRISM is shown in Fig. 4 and illustrated as follows:

#### 3.1.1. STAGE 1: DECOMPOSITION & INITIALIZATION

Instead of treating the task vector $\tau_{\text{lrm}}$ as a monolithic update, we decompose it into orthogonal subspaces via SVD to conduct merging at the subspace level. Consider the $l$-th layer, a linear layer with a dimension $d_{\text{out}} \times d_{\text{in}}$. We apply SVD to its corresponding reasoning task vector:

$$\mathbf{U}^{(l)}, \mathbf{S}^{(l)}, \mathbf{V}^{(l)\top} = \text{SVD}(\tau_{\text{lrm}}^l), \tag{6}$$

where $\mathbf{U} \in \mathbb{R}^{d_{\text{out}} \times r}$ and $\mathbf{V} \in \mathbb{R}^{r \times d_{\text{in}}}$ represent the left and right singular vectors, and $\mathbf{S} = \text{diag}(\sigma_1, \ldots, \sigma_r) \in \mathbb{R}^{r \times r}$ represents the singular values, indicating the merging coefficient of each subspace. We initialize the merged layers with these decomposed matrices. Crucially, to preserve the semantic direction of the reasoning task, we freeze the orthogonal bases $\mathbf{U}$ and $\mathbf{V}$ and the base intensity $\mathbf{S}$ during the entire training phase.

To achieve fine-grained control over the reasoning injection, we further introduce a lightweight, learnable parameter $\mathbf{g}^l \in \mathbb{R}^r$ for the $l$-th layer, initialized to zero. Acting as an adaptive merging coefficient modulator, $\mathbf{g}$ dynamically adjusts the intensity of each subspace to be merged. Note that, following IP-Merging (Hu et al., 2025), to stabilize visual performance, we fix the merging coefficient of the

VLM to 1.0. Therefore, the merged layer is formulated as:

$$
\begin{aligned}
\theta_{\text{merged}}^l &= \theta_{\text{vlm}}^l + \Delta\theta_{\text{eff}}^l \\
&= \theta_{\text{vlm}}^l + \lambda_{\text{lrm}} \cdot \mathbf{U}^{(l)} \mathbf{S}_{\text{eff}}^{(l)} \mathbf{V}^{(l)\top} \\
&= \theta_{\text{vlm}}^l + \lambda_{\text{lrm}} \cdot \mathbf{U}^{(l)} \left( \text{Sigmoid}(\mathbf{g}^l) \odot \mathbf{S}^{(l)} \right) \mathbf{V}^{(l)\top},
\end{aligned} \tag{7}
$$

where $\Delta\theta_{\text{eff}}^l$ is the modulated subspace of the reasoning task vector, $\lambda_{\text{lrm}}$ is the merging coefficient for reasoning, and $\odot$ denotes element-wise multiplication. After going through the Sigmoid activation function, the scaling factor of each subspace lies within $(0, 1)$. Herein, $\mathbf{S}_{\text{eff}} = \text{Sigmoid}(\mathbf{g}^l) \odot \mathbf{S}^{(l)}$ represents the gated effective singular values. Regarding the subspaces of reasoning task vectors as reasoning priors, we keep the singular vectors frozen and only train the gating matrix. Therefore, the number of trainable parameters of FRISM is quite small, making FRISM extremely parameter-efficient.

#### 3.1.2. STAGE 2: INJECTION & TRAINING

To maximize reasoning injection intensity through merging and minimize visual perception loss, we apply a label-free self-distillation framework and conduct dual-objective optimization as follows:

**Visual Preservation.** Specifically, we treat the original base VLM $\theta_{\text{vlm}}$ as the frozen teacher and the merged model $\theta_{\text{vlrm}}$ (parameterized by the learnable gates $g$) as the student. To ensure that the injection of reasoning subspaces does not disrupt the post-training vision alignment, we minimize the Kullback-Leibler (KL) divergence between the output probability distributions of the two models on a calibration dataset $\mathcal{D}$. In our experiments, we apply VizWiz (Gurari et al., 2018), a real-world visual question answering (VQA) dataset, for calibration. This objective enforces consistency in the model's predictive behavior for visual inputs, effectively penalizing gating configurations that lead to significant semantic drift. The distillation loss is formulated as:

$$\mathcal{L}_{\text{distill}} = \mathbb{E}_{x \sim \mathcal{D}} \left[ \text{KLD} \left( P(\cdot|x; \theta_{\text{vlm}}) \parallel P(\cdot|x; \theta_{\text{vlrm}}) \right) \right], \tag{8}$$

where $P(\cdot|x; \theta)$ denotes the output probability distribution given input $x$. This objective encourages the merged model to retain the robust visual understanding capabilities of the VLM before merging.

**Reasoning Maximization.** Simply minimizing $\mathcal{L}_{\text{distill}}$ would lead to a trivial solution where $\mathbf{g} \to -\infty$, which means no subspace from the LRM is merged to the VLM. To encourage the absorption of reasoning capabilities, since VL reasoning datasets are not accessible in our setting, we apply the subspaces of LRM task vectors as reasoning priors and introduce a maximization term for the effective magni-

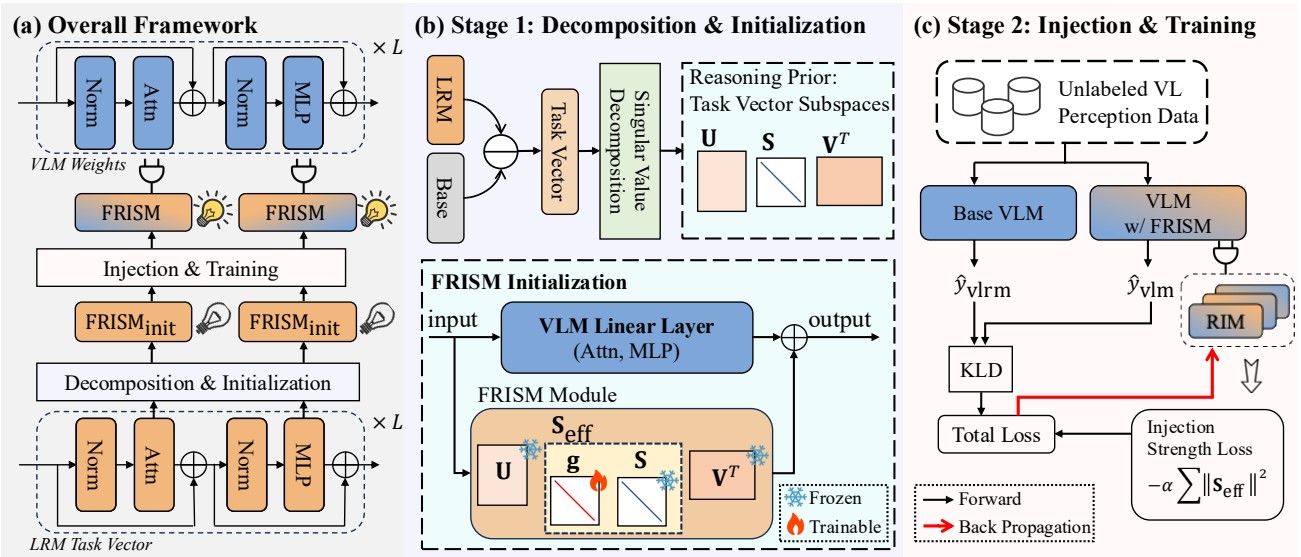

*Figure 4.* FRISM Framework. (a) The overall framework. FRISM transfers the reasoning capabilities from LRMs to VLMs. FRISM can be divided into two stages. (b) Stage 1: Decomposition and Initialization. (c) Stage 2: Injection and Training.

tude of each subspace:

$$\mathcal{L}_{\text{inject}} = -\sum_{l=1}^{L} \|\mathbf{S}_{\text{eff}}^{(l)}\|^2 = -\sum_{l=1}^{L} \|\text{Sigmoid}(\mathbf{g}^{(l)}) \odot \mathbf{S}^{(l)}\|^2. \tag{9}$$

This term encourages enlarging the merging coefficients of reasoning subspaces defined by the LRM, thus improving the overall reasoning capabilities of the model. Practically, because the scales of different models vary greatly, the injection loss $\mathcal{L}_{\text{inject}}$ is normalized before training; see Appendix H. Please check Appendix F.2 for the detailed analysis of the loss terms.

**Total Objective.** The final loss function balances these two terms:

$$\mathcal{L} = \mathcal{L}_{\text{distill}} + \alpha \cdot \mathcal{L}_{\text{inject}}, \tag{10}$$

where $\alpha$ is a hyperparameter that controls the injection strength. This optimization process automatically discovers the subspaces that yield the optimal trade-off between reasoning injection and visual preservation. It effectively acts as a filter of subspaces: it retains reasoning components that are orthogonal to visual perception capabilities while suppressing those that cause high visual degradation (high KL divergence). It should be noted that during the training process, only the learnable gating parameters are trainable, while the original VLM weights and the decomposed singular vectors remain frozen. This ensures that the method remains highly parameter-efficient and preserves the foundational visual capabilities of the base model. The algorithm flow of the proposed FRISM is shown in the Appendix A. More technique details of FRISM is in the Appendix H.

### 3.2. Theoretical Analysis

We provide a theoretical analysis to demonstrate that FRISM acts as a filter that selectively amplifies reasoning-beneficial subspaces and suppresses vision-degrading ones. We formulate the merging process as a constrained optimization problem, where the goal is to maximize reasoning gain while being bounded by visual degradation.

Let $\mathcal{L}_{\text{vis}}(\cdot)$ denote the visual loss function. Assuming that the base VLM $\theta_{\text{vlm}}$ has converged to a local minimum. Approximating via Taylor expansion, the visual degradation $\delta_{\text{vis}}$ caused by a parameter update $\Delta\theta$ is determined by:

$$\mathcal{L}_{\text{vis}}(\theta_{\text{vlm}} + \Delta\theta) \approx \frac{1}{2}\Delta\theta^\top \mathbf{H}\Delta\theta, \tag{11}$$

where $\mathbf{H} = \nabla^2 \mathcal{L}_{\text{vis}}(\theta_{\text{vlm}})$ is the hessian matrix. In standard layer-wise merging, if the task vector aligns with the high-eigenvalue directions of $\mathbf{H}$ (high curvature), the visual degradation can become severe. Since labeled VL reasoning datasets are not accessible in our setting, we use the norm of the task vector as a proxy for reasoning injection strength. Let $\lambda_i = \lambda_{\text{lrm}} \cdot \text{Sigmoid}(\mathbf{g}^{(l)})$ and $B_i = \mathbf{U}[:, i]\mathbf{S}[i]\mathbf{V}[i, :]$, the total loss becomes:

$$\mathcal{L} = \mathcal{L}_{\text{vis}} + \mathcal{L}_{\text{reason}}$$

$$= \frac{1}{2}\left(\sum_{i=1}^{r} \lambda_i B_i\right)^\top \mathbf{H}\left(\sum_{j=1}^{r} \lambda_j B_j\right) - \alpha \sum_{i=1}^{r} \lambda_i^2 \|B_i\|_F^2, \tag{12}$$

Based on the assumption that different SVD subspaces are approximately decoupled with respect to the visual loss curvature, i.e., $\text{Tr}(B_i^\top \mathbf{H} B_j) \approx 0$ for $i \neq j$, $\mathcal{L}$ with respect

to $\lambda_i$ can be written as:

$$\frac{\partial \mathcal{L}}{\partial \lambda_i} \approx \left(h_i - 2\alpha \|B_i\|_F^2\right) \lambda_i, \tag{13}$$

where $h_i = \text{Tr}(B_i^\top \mathbf{H} B_i)$. Depending on whether $h_i - 2\alpha \|B_i\|_F^2$ is positive or negative for the $i$-th subspace, the adaptive merging coefficient is suppressed or enlarged.

A detailed theoretical analysis is in the Appendix F.

## 4. Experiments

### 4.1. Experimental Setup

**Baseline Methods.** We compare the proposed FRISM with several baseline methods, including: (1) Base VLM, i.e., the vision language model before merging; (2) Task Arithmetic (TA) (Ilharco et al., 2023); (3) Ties-Merging (Yadav et al., 2023); (4) DARE (Yu et al., 2024); and (5) IP-Merging (Hu et al., 2025).

Please check Appendix C.2 for detailed information about baselines and hyper-parameter selection of existing merging methods.

**Datasets.** The performance of each model is evaluated by several multimodal tasks, including six vision-language reasoning tasks and three visual perception tasks. The vision-language reasoning tasks include: (1) MathVista (Lu et al., 2024a), (2) MathVision (Wang et al., 2024a), (3) Math-Verse (Zhang et al., 2024), (4) MMMU (Yue et al., 2024), (5) R1-OneVision (Yang et al., 2025a), and (6) MMStar (Chen et al., 2024a). The visual perception tasks include: (1) TextVQA (Singh et al., 2019), (2) POPE (Li et al., 2023b), (3) SeedBench (Li et al., 2023a). Please check Appendix C.3 for more details about the datasets. The evaluation details are declared in Appendix C.4. All the model IDs can be found in Appendix C.1.

### 4.2. Experimental Results

#### 4.2.1. EXPERIMENTS ON QWEN2.5-VL SERIES MODELS

**Settings.** To validate the scalability and versatility of our proposed method, we conduct experiments across three different model scales within the Qwen2.5-VL (Bai et al., 2025) family. Specifically, we construct three merging pairs: (1) *3B Scale:* merging Qwen2.5-VL-3B-Instruct with the reasoning model SmallThinker-3B (Song et al., 2025), an efficient reasoning model designed for edge devices; (2) *7B Scale:* merging Qwen2.5-VL-7B-Instruct with DeepSeek-R1-Distill-Qwen-7B (Guo et al., 2025), a distilled reasoning model with long CoT capabilities; and (3) *32B Scale:* merging the large-scale Qwen2.5-VL-32B-Instruct with QwQ-32B (Team, 2025), a long CoT model with careful reflection and self-questioning capabilities. All the VLMs and LRMs are post-trained from the Qwen2.5 (Yang et al., 2024a; Team,

2024) series language model.

**Results.** The quantitative results are presented in Tab. 1. As shown, FRISM achieves consistent and significant improvements over existing merging methods across all model scales (3B, 7B, and 32B), demonstrating exceptional scalability and robustness. On the 3B scale, FRISM surpasses the best baseline method and the base VLM by 1.8 points, while maintaining the visual perception score. Notably, the performance improvement on MathVista reaches 4.5 points. This advantage is further amplified at the 7B scale, where FRISM obtains the highest reasoning accuracy of 49.4, surpassing the base VLM by 2.0 points and outperforms all the baseline methods. It should be noted that the VL reasoning performance of FRISM on the 7B scale is close to ThinkLite-VL-7B (Wang et al., 2025b), which has undergone high-cost post-training. On VL perception tasks, existing methods show obvious performance degradation while FRISM shows a slight improvement compared to the base VLM. On the 32B scale, FRISM shows a more significant improvement of 2.4 points. Notably, on MathVision, our method achieves an improvement of 4.4 points. On VL perception tasks, the performance drop of FRISM is still the least among all the methods.

**Keyword Analysis.** To further validate the reasoning and vision capabilities of different models, we analyze the reasoning-related and vision-related keyword tokens generated by the models merged through different methods when merging Qwen2.5-VL-32B-Instruct and QwQ-32B on the MathVision dataset. Because QwQ-32B is a model capable of self-reflection and multi-step logical reasoning, we verify the strength of the reflective capabilities injected into different models by counting the number of reasoning-related reflection tokens in Fig. 5. It can be seen that FRISM generates the most reflection tokens compared to the base model and baseline methods, demonstrating that the reasoning capabilities are injected most effectively compared to other methods. Meanwhile, the number of vision-related tokens generated by FRISM is also improved compared to the base VLM, validating that FRISM preserves visual capabilities during merging. Please refer to Appendix I for more information. Visualization results are in Appendix G.

#### 4.2.2. EXPERIMENTS ON MORE TYPES OF MODELS

**Settings.** To further verify the applicability of FRISM, we conduct additional experiments on different kinds of models, including: (1) merging ThinkLite-VL-7B (Wang et al., 2025b) and DeepSeek-R1-distilled-Qwen-7B (Guo et al., 2025); (2) merging Qwen2.5-VL-7B-Instruct (Bai et al., 2025) and OpenThinker3-7B (Guha et al., 2025); (3) merging InternVL3-8B (Zhu et al., 2025) and DeepSeek-R1-distilled-Qwen-7B (Guo et al., 2025).

**Results.** The results are in Tab. 2. As shown, FRISM

*Table 1.* Comparison with other model merging approaches on the VL Reasoning benchmarks and VL perception benchmarks when merging Qwen2.5-VL series models with language reasoning models of sizes from 3B to 32B. **Bold** represents the best performance and underline represents the second-best performance among different merging methods.

| Methods | VL Reasoning Benchmarks | | | | | | | VL Perception Benchmarks | | | |
|---|---|---|---|---|---|---|---|---|---|---|---|
| | MVista | MVerse | MVision | MMMU | R1-OV | MMStar | Avg | TextVQA | POPE | Seed | Avg |
| *Qwen2.5-VL-3B-Instruct Merge SmallThinker-3B* | | | | | | | | | | | |
| Base | 40.5 | 25.5 | 20.6 | 44.0 | 28.0 | 40.5 | 33.2 | 79.3 | 86.2 | 73.7 | 79.7 |
| TA ($\lambda = 0.1$) | 41.5 | 26.7 | 20.6 | 39.0 | 28.8 | 40.7 | 32.9 | **79.2** | 86.3 | **73.8** | **79.8** |
| TA ($\lambda = 0.15$) | 38.5 | 27.9 | 19.7 | 39.2 | **29.0** | 43.2 | 32.9 | 79.1 | **86.6** | 73.7 | **79.8** |
| TA ($\lambda = 0.2$) | 41.4 | 28.0 | 21.2 | 37.8 | 27.5 | 42.1 | 33.0 | 79.0 | **86.6** | **73.8** | **79.8** |
| Ties-Merging | 41.8 | 17.8 | 17.4 | **45.0** | 25.9 | 41.4 | 31.6 | 76.4 | 86.5 | 73.6 | 77.0 |
| DARE | 40.7 | 27.1 | 20.5 | 41.0 | 25.8 | 40.4 | 32.6 | **79.2** | 86.2 | **73.8** | 79.7 |
| IP-Merging ($T = 0.2$) | 21.8 | 15.8 | 8.6 | 25.6 | 14.1 | 27.0 | 18.8 | 70.6 | 88.2 | 72.7 | 77.2 |
| IP-Merging ($T = 0.3$) | 41.1 | 24.1 | 18.5 | 42.0 | 27.0 | 40.7 | 32.2 | 76.9 | 81.1 | 72.9 | 77.0 |
| IP-Merging ($T = 0.4$) | 41.1 | 24.1 | 18.5 | 42.0 | 27.0 | 40.7 | 32.2 | 76.9 | 81.1 | 72.9 | 77.0 |
| FRISM (Ours) | **45.0** | **28.3** | **21.4** | 42.8 | 28.9 | **43.4** | **35.0**$^{\uparrow 1.8}$ | **79.2** | 86.2 | **73.8** | 79.7 |
| *Qwen2.5-VL-7B-Instruct Merge DeepSeek-R1-Distill-Qwen-7B* | | | | | | | | | | | |
| Base | 68.1 | 41.2 | 25.6 | 55.1 | 34.6 | 60.1 | 47.4 | 85.5 | 86.4 | 77.0 | 82.9 |
| TA ($\lambda = 0.1$) | 65.4 | 37.8 | 24.2 | 53.7 | 36.6 | 60.7 | 46.4 | 83.9 | 86.8 | 76.5 | 82.4 |
| TA ($\lambda = 0.15$) | 32.9 | 17.1 | 10.7 | 38.3 | 22.0 | 40.3 | 26.9 | 69.1 | 73.9 | 63.9 | 69.0 |
| TA ($\lambda = 0.2$) | 28.2 | 14.8 | 9.2 | 34.6 | 21.2 | 34.6 | 23.8 | 68.0 | 73.4 | 61.0 | 67.5 |
| Ties-Merging | 61.9 | 38.8 | 23.5 | 53.0 | 35.7 | 58.6 | 45.3 | 77.9 | 84.2 | 74.6 | 78.9 |
| DARE | 68.4 | 41.1 | 25.4 | 56.4 | 34.6 | 60.8 | 47.8 | 84.9 | 85.6 | 76.9 | 82.5 |
| IP-Merging ($T = 0.2$) | 65.9 | 38.7 | 24.9 | 55.1 | 33.9 | 58.7 | 46.2 | 81.6 | 84.9 | **77.0** | 81.2 |
| IP-Merging ($T = 0.3$) | 68.8 | 40.6 | 25.0 | 54.7 | 35.9 | 60.1 | 47.5 | 82.0 | 86.4 | **77.0** | 81.8 |
| IP-Merging ($T = 0.4$) | 67.2 | 40.8 | 25.5 | 55.9 | 36.0 | 60.7 | 47.7 | 82.9 | **87.1** | 76.9 | 82.3 |
| FRISM (Ours) | **70.3** | **41.8** | **26.4** | **57.8** | **37.5** | **62.4** | **49.4**$^{\uparrow 2.0}$ | **85.0** | **87.1** | 76.9 | **83.0** |
| *Qwen2.5-VL-32B-Instruct Merge QwQ-32B* | | | | | | | | | | | |
| Base | 77.7 | 47.0 | 38.2 | 65.2 | 47.5 | 67.9 | 57.3 | 79.8 | 86.5 | 69.6 | 78.6 |
| TA | 78.1 | 48.2 | 40.3 | **67.6** | 50.0 | 67.7 | 58.7 | **79.9** | 86.5 | **68.3** | 78.2 |
| Ties-Merging | 74.8 | 42.8 | 35.2 | 63.9 | 42.1 | 65.3 | 54.0 | 78.3 | 86.3 | 58.6 | 74.4 |
| IP-Merging | 75.5 | 45.2 | 38.4 | 65.9 | 48.3 | **67.9** | 56.9 | 75.8 | **87.9** | 67.4 | 77.1 |
| FRISM (Ours) | **79.8** | **49.3** | **42.6** | 67.0 | **51.9** | 67.8 | **59.7**$^{\uparrow 2.4}$ | **79.9** | 87.0 | 67.9 | **78.3** |

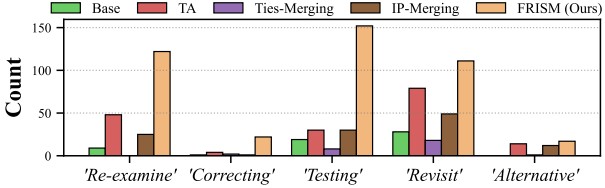

*Figure 5.* Count of self-reflection tokens on MathVision dataset of different merging methods.

demonstrates consistent and significant improvements, further validating its versatility and robustness. When merging ThinkLite-VL-7B with DeepSeek-R1-distilled-Qwen-7B, the VLM undergoes post-training for visual alignment and reasoning capabilities compared to pre-trained models. Consequently, the parameter differences between the VLM and LRM are significant under this setting, resulting in the failure of existing methods. None of the existing methods obtain performance gains through merging, while FRISM achieves performance gains on nearly all tasks and retains

visual performance on VL perception tasks. When merging Qwen2.5-VL-7B-Instruct with OpenThinker3-7B, FRISM achieves both the highest reasoning score and VL perception performance. In this setting, IP-Merging suffers from performance collapse under $T = 0.2$ and $T = 0.3$. This issue is due to the exceptionally large merging coefficients calculated by IP-Merging and is discussed in detail in Appendix E. When applied to the InternVL3-8B architecture, FRISM also achieves remarkable performance on both VL reasoning and perception tasks. We also conduct experiments on Llama-based models in Appendix D.2.

### 4.3. Efficiency Analysis

In Tab. 3, we compare the proposed FRISM with the base VLM and DRIFT (Huang et al., 2025), an efficient post-training method through supervised finetuning on a self-constructed VLM reasoning dataset, in terms of training costs and reasoning performance. Note that we reproduce the performance of DRIFT in our experimental settings. Additionally, to further reduce the training costs of the pro-

*Table 2.* Comparison with other model merging approaches when merging ThinkLite-VL-7B with DeepSeek-R1-distilled-Qwen-7B, merging Qwen2.5-VL-7B-Instruct with OpenThinker3-7B, and merging InternVL-3-8B model with DeepSeek-R1-distilled-Qwen-7B.

| Methods | VL Reasoning Benchmarks | | | | | | | VL Perception Benchmarks | | | |
|---|---|---|---|---|---|---|---|---|---|---|---|
| | MVista | MVerse | MVision | MMMU | R1-OV | MMStar | **Avg** | TextVQA | POPE | Seed | **Avg** |
| *ThinkLite-VL-7B Merge DeepSeek-R1-distilled-Qwen-7B* | | | | | | | | | | | |
| Base | 73.3 | 42.9 | 27.2 | 56.8 | 36.4 | 61.7 | 49.7 | 85.5 | 86.5 | 76.7 | 82.9 |
| TA ($\lambda = 0.1$) | 71.4 | 40.0 | 25.7 | 56.6 | 37.8 | 62.2 | 49.0 | 84.4 | 86.6 | 76.6 | 82.5 |
| TA ($\lambda = 0.15$) | 34.7 | 19.8 | 13.2 | 38.7 | 23.7 | 41.7 | 28.6 | 68.6 | 72.6 | 61.9 | 67.7 |
| TA ($\lambda = 0.2$) | 31.7 | 14.9 | 10.8 | 39.2 | 19.0 | 39.5 | 25.9 | 62.8 | 62.6 | 70.7 | 65.4 |
| Ties-Merging | 67.4 | 36.8 | 24.3 | 52.6 | 35.8 | 59.3 | 46.0 | 74.3 | 84.3 | 65.4 | 74.7 |
| DARE | 73.1 | 41.2 | 25.9 | 56.1 | 35.9 | 63.0 | 49.2 | **85.4** | **86.7** | 76.7 | **82.9** |
| IP-Merging ($T = 0.2$) | 72.2 | 40.5 | 24.9 | 54.9 | 33.8 | 61.9 | 48.0 | 82.6 | 86.0 | 76.7 | 81.7 |
| IP-Merging ($T = 0.3$) | 73.2 | 41.4 | 27.5 | 56.5 | 34.6 | 62.9 | 49.3 | 82.5 | 86.8 | **76.8** | 82.0 |
| IP-Merging ($T = 0.4$) | 73.4 | 41.8 | 27.1 | 56.9 | 36.3 | 61.5 | 49.5 | 83.2 | 87.2 | 76.7 | 82.3 |
| FRISM (Ours) | **74.0** | **42.8** | **28.1** | **58.7** | **38.4** | **63.1** | **50.9**[↑1.2] | 85.4 | 86.6 | **76.8** | 82.9 |
| *Qwen2.5-VL-7B-Instruct Merge OpenThinker3-7B* | | | | | | | | | | | |
| Base | 68.1 | 41.2 | 25.6 | 55.1 | 34.6 | 60.1 | 47.4 | 85.5 | 86.4 | 77.0 | 82.9 |
| TA ($\lambda = 0.1$) | 67.6 | 40.3 | 25.9 | **56.4** | 34.7 | 58.5 | 47.2 | 85.0 | **86.5** | 76.8 | 82.8 |
| TA ($\lambda = 0.15$) | 67.2 | 38.2 | 23.6 | 53.4 | 34.9 | 59.2 | 46.1 | 84.1 | 86.4 | 76.4 | 82.3 |
| TA ($\lambda = 0.2$) | 63.5 | 38.4 | 21.9 | 54.6 | 35.2 | 58.1 | 45.3 | 82.9 | 86.4 | 75.1 | 81.5 |
| Ties-Merging | 69.0 | 41.8 | 25.5 | 55.2 | 33.9 | 59.8 | 47.5 | 83.8 | 86.4 | 76.9 | 82.4 |
| DARE | 68.9 | 41.9 | **27.2** | 54.5 | 34.8 | 60.1 | 47.9 | 85.4 | 86.4 | 76.9 | 82.9 |
| IP-Merging | 67.4 | 39.1 | 25.1 | 55.9 | 34.6 | **61.4** | 47.3 | 84.8 | 86.1 | **77.0** | 82.6 |
| FRISM (Ours) | **69.9** | **42.1** | 27.2 | 56.3 | **37.5** | 61.3 | **49.0**[↑1.6] | **85.5** | **86.5** | **77.0** | **83.0** |
| *InternVL3-8B Merge DeepSeek-R1-Distill-Qwen-7B* | | | | | | | | | | | |
| Base | 72.0 | 42.3 | 29.5 | 60.7 | 39.5 | 68.9 | 52.2 | 82.2 | 90.3 | 77.2 | 83.2 |
| TA | 68.2 | 39.2 | 26.4 | 56.1 | 39.7 | 65.8 | 49.2 | 81.8 | 90.0 | **77.6** | 83.1 |
| Ties-Merging | 66.0 | 38.6 | 25.5 | 55.1 | 37.5 | 62.6 | 47.6 | 80.3 | 88.7 | 77.2 | 82.1 |
| DARE | 69.6 | 40.2 | 28.9 | 57.0 | 39.8 | 67.2 | 50.5 | 82.0 | 90.7 | 77.4 | **83.4** |
| IP-Merging | 72.3 | 42.1 | 30.3 | 59.8 | **42.1** | 68.9 | 52.6 | 82.1 | 90.2 | 77.2 | 83.2 |
| FRISM (Ours) | **74.0** | **42.6** | **30.7** | **61.0** | 41.2 | **69.3** | **53.2**[↑1.0] | **82.2** | 90.3 | 77.4 | 83.3 |

*Table 3.* Comparison of computational resource consumption and performance between ours (both full-rank and low-rank) and DRIFT (Huang et al., 2025).

| Methods | Training Costs | | Performance | | | | |
|---|---|---|---|---|---|---|---|
| | Trainable Params (M) | Training Time (h) | MVista | MVerse | MVision | R1-OV | Avg |
| Qwen2.5-VL-7B-Inst | - | - | 68.1 | 41.2 | 25.6 | 34.6 | 42.4 |
| DRIFT | ∼7000 | ∼2 | **70.4** | **41.8** | 25.7 | 34.9 | 43.2 |
| Ours | 0.53 | 0.49 | 70.3 | **41.8** | **26.4** | **37.5** | **44.0** |
| Ours_lowrank | 0.05 | 0.43 | 69.3 | 41.6 | 26.3 | 36.9 | 43.5 |

*Table 4.* Ablation on the decomposition process and the injection loss of FRISM.

| Methods | MVista | MVerse | MVision | MMMU | R1-OV | MMStar | Avg |
|---|---|---|---|---|---|---|---|
| Qwen2.5-VL-7B-Inst | 68.1 | 41.2 | 25.6 | 55.1 | 34.6 | 60.1 | 47.4 |
| FRISM (w/o SVD) | 67.5 | 39.9 | 26.7 | 55.8 | 37.3 | 60.1 | 47.9 |
| FRISM (w/o $\mathcal{L}_{\text{inject}}$) | 68.3 | 41.5 | 25.6 | 55.2 | 35.4 | 60.7 | 47.8 |
| FRISM (Ours) | **70.3** | **41.8** | **26.4** | **57.8** | **37.5** | **62.4** | **49.4** |

*Table 5.* Ablation on the gate range of FRISM.

| Methods | MVista | MMStar | R1-OV | TextVQA | OCRBench |
|---|---|---|---|---|---|
| Qwen2.5-VL-7B-Inst | 68.1 | 60.1 | 34.6 | 85.5 | 87.7 |
| FRISM (Softplus) | 68.8 | 60.3 | 36.0 | 84.2 | 87.4 |
| FRISM (Sigmoid, Ours) | **70.3** | **62.4** | **37.5** | **85.0** | **87.9** |

posed FRISM, we perform a truncation on the singular values and singular vectors of the task vectors. Specifically, we retain only the top-256 singular values and their corresponding singular vectors in matrices $\mathbf{U}$, $\mathbf{S}$, and $\mathbf{V}$, thus reducing the trainable parameters and training time costs. The results show that FRISM is a highly efficient framework, and its efficiency can be further maximized under the low-rank setting. Remarkably, despite this minimal overhead, our method achieves reasoning performance comparable to and even surpassing that of data-dependent post-training DRIFT, demonstrating that FRISM effectively bridges the performance gap with resource-heavy post-training approaches

while offering significantly higher computational efficiency.

### 4.4. Ablation Studies

**Ablation on the decomposition process.** To further demonstrate the effectiveness of merging in the subspace level, in Tab. 4, we conduct an ablation study on the SVD process. Specifically, we compare our proposed FRISM with a

*Table 6.* Ablation on the decomposition strategy of FRISM.

| Methods | MVista | MMStar | R1-OV | TextVQA | OCRBench |
|---|---|---|---|---|---|
| Qwen2.5-VL-3B-Inst | 40.5 | 40.5 | 28.0 | 79.3 | 82.6 |
| FRISM (Rand-Mask) | 41.3 | 41.9 | 27.8 | 79.0 | 82.3 |
| FRISM (SVD, Ours) | **45.0** | **43.4** | **28.9** | **79.2** | **82.7** |

variant that does not include the SVD process, denoted as FRISM (w/o SVD) in Tab. 4, a degenerated variant of adaptive layer-wise merging. Specifically, instead of fine-grained subspace modulation, we assign a single learnable gating scalar $\mathbf{g}^{(l)} \in \mathbb{R}$ to the entire task vector of the LRM. For this variant, the loss function and the training process follow FRISM. This variant yields a marginal improvement over the base VLM on VL reasoning benchmarks. In contrast, our FRISM significantly outperforms the variant without SVD, empirically demonstrating the advantages of merging in the subspace level over layer-wise merging.

**Ablation on the injection loss.** To empirically demonstrate the effectiveness of the reasoning maximization loss term, we conduct ablation studies on $\mathcal{L}_{\text{inject}}$. The results after training without $\mathcal{L}_{\text{inject}}$ are shown in Tab. 4. The performance is close to the base VLM, indicating that after removing $\mathcal{L}_{\text{inject}}$ during FRISM training, the model tends to degrade to the base VLM. This demonstrates the effectiveness of the reasoning maximization term in our loss function.

**Ablation on the gate range.** We design the per-subspace Sigmoid gate as a bounded filter because post-hoc reasoning injection is fragile: unrestricted coefficients may cause visual drift, destabilize the merged model, and allow some subspaces to dominate, disrupting the model's original properties. The overall injection magnitude is controlled by $\lambda_{\text{lrm}}$ and the gated singular values, while the sigmoid gate can provide conservative, fine-grained selection of reasoning subspaces under the visual-preservation constraint. An ablation study on the gate range is shown in Tab. 5. It can be seen that FRISM with the Sigmoid gate shows better performance compared to the variant with the Softplus gate.

**Ablation on decomposition strategy.** To validate the effectiveness of SVD compared to other decomposition methods, we apply random masks to LRM task vectors for decomposition. The results based on Qwen2.5-3B series models are shown in Tab. 6. It can be seen that FRISM (w/ SVD) consistently outperforms the variant with a random mask, further demonstrating the effectiveness of SVD. Additionally, compared to mask-based decomposition, subspaces are low-rank and require less GPU memory during training.

### 4.5. More Experimental Results

In Appendix D, we show additional experimental results, including results on more visual perception benchmarks and comparisons with more reasoning VLMs. We also present case studies in Appendix J and K.

## 5. Conclusions

In this paper, we propose FRISM, a fine-grained adaptive model merging framework designed to resolve the conflict between injecting reasoning capabilities and preserving visual perception in VLMs through model merging. Unlike existing layer-wise merging methods that often compromise one capability for the other, FRISM operates at the subspace level via SVD, enabling the precise identification and integration of reasoning-dominant components. Furthermore, we introduce a label-free self-distillation strategy with dual-objective optimization to adaptively modulate injection strength without relying on scarce multimodal reasoning datasets. Extensive experiments across various model structures and scales on multiple benchmarks demonstrate that FRISM consistently achieves impressive performance, effectively bridging the gap between powerful LRMs and VLMs. FRISM offers an efficient, plug-and-play solution for enhancing vision-language reasoning while maintaining visual robustness.

## Acknowledgements

This work is supported by National Key R&D Program of China (No. 2026YFE0101200), and Shanghai Natural Science Foundation (No. 23ZR1402900). The computations in this research were performed using the CFFF platform of Fudan University.

## Impact Statement

This paper presents work whose goal is to advance the field of Machine Learning. There are many potential societal consequences of our work, none of which we feel must be specifically highlighted here.

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

# Supplementary Materials for FRISM

## A. Algorithm Flow of FRISM

We present the algorithm flow of the proposed FRISM in Algorithm 1.

## B. Configuration of Fig. 3

In Fig. 3, we present the performance on R1-OneVision when merging Qwen2.5-VL-7B-Instruct and the subspaces of the DeepSeek-R1-Distill-Qwen-7B task vector. Due to the fact that subspaces of different ranks have different singular values. Therefore, the magnitudes of different subspaces differ. Before merging, the subspaces of the task vectors are aligned to the original task vector based on L2 Norm. Specifically, when merging the rank-$n$ subspace of LRM at the coefficient $\lambda$, the merging process can be written as:

$$
\begin{aligned}
\mathbf{U}, \mathbf{S}, \mathbf{V} &= \mathrm{SVD}\left(\Delta\theta_{\mathrm{lrm}}\right); \\
\Delta\theta_{\mathrm{lrm}}^{\mathrm{rank}=n} &= \mathbf{U}[:,n]\mathbf{S}[n,n]\mathbf{V}[n,:]; \\
\theta_{\mathrm{merged}}\left(n\right) &= \theta_{\mathrm{vlm}} + \Delta\theta_{\mathrm{lrm}}^{\mathrm{rank}=n} \cdot \frac{\|\Delta\theta_{\mathrm{lrm}}\|^2}{\|\Delta\theta_{\mathrm{lrm}}^{\mathrm{rank}=n}\|^2} \cdot \lambda.
\end{aligned}
\tag{A1}
$$

## C. Detailed Experimental Settings

### C.1. Model IDs

In Table A1, we present the Huggingface (Wolf et al., 2019) IDs of the models used in our experiments, including VLMs and LRMs.

*Table A1.* Model-ID of the models, including VLMs and LRMs, for the examined tasks.

| Model Name | | Huggingface ID |
|---|---|---|
| **VLMs** | Qwen2.5-VL-7B-Instruct | Qwen/Qwen2.5-VL-7B-Instruct |
| | ThinkLite-VL-7B | russwang/ThinkLite-VL-7B |
| | Qwen2.5-VL-3B-Instruct | Qwen/Qwen2.5-VL-3B-Instruct |
| | LLaVA-next-8b | lmms-lab/llama3-llava-next-8b |
| | Qwen2.5-VL-32B-Instruct | Qwen/Qwen2.5-VL-32B-Instruct |
| | DRIFT-VL-7B | ChaoHuangCS/DRIFT-VL-7B |
| | OpenVLThinker-7B | ydeng9/OpenVLThinker-7B |
| | VLAA-Thinker-Qwen2.5VL-7B | UCSC-VLAA/VLAA-Thinker-Qwen2.5VL-7B |
| **LRMs** | DeepSeek-R1-Distill-Qwen-7B | deepseek-ai/DeepSeek-R1-Distill-Qwen-7B |
| | OpenThinker3-7B | open-thoughts/OpenThinker3-7B |
| | SmallThinker-3B | Tiiny/SmallThinker-3B-Preview |
| | DeepThought-8B | ruliad/deepthought-8b-llama-v0.01-alpha |
| | QwQ-32B | Qwen/QwQ-32B |

### C.2. Baseline Methods

**Task Arithmetic** (Ilharco et al., 2023) defines task vectors as the difference between finetuned model weights

---

**Algorithm 1** FRISM Algorithm Flow

**Input:** VLM $\theta_{\mathrm{vlm}}$, LRM $\theta_{\mathrm{lrm}}$, Base Model $\theta_{\mathrm{base}}$.
**Hyper-Params:** Merging Coefficient $\lambda_{\mathrm{lrm}}$, Learning Rate $\eta$, Injection Coefficient $\alpha$.
**Output:** VLM with reasoning capabilities $\theta_{\mathrm{vlrm}}$.
▷ Stage 1: Initialization.
**for** *each linear layer* $l \in \{1, \dots, L\}$ **do**
   ▷ Compute task vectors.
   $\tau_{\mathrm{lrm}}^l \leftarrow \theta_{\mathrm{lrm}}^l - \theta_{\mathrm{base}}^l; \qquad \tau_{\mathrm{vlm}}^l \leftarrow \theta_{\mathrm{vlm}}^l - \theta_{\mathrm{base}}^l$
   ▷ Perform SVD.
   $\mathbf{U}^{(l)}, \mathbf{S}^{(l)}, \mathbf{V}^{(l)\top} \leftarrow \mathrm{SVD}(\tau_{\mathrm{lrm}}^l)$
   ▷ Initialize trainable parameters and model.
   $\mathbf{g}^{(l)} \leftarrow \mathbf{0}$
   $\theta_{\mathrm{vlrm}}^l \leftarrow \theta_{\mathrm{vlm}}^l + \lambda_{\mathrm{lrm}} \cdot \mathbf{U}^{(l)}\left(\mathrm{Sigmoid}(\mathbf{g}^{(l)}) \odot \mathbf{S}^{(l)}\right)\mathbf{V}^{(l)\top}$
**end**
▷ Stage 2: Training.
**while** *not converged* **do**
   Sample batch $\mathbf{x} \sim \mathcal{D}$
   ▷ Base VLM and injected VLRM inference.
   $\hat{\mathbf{y}}_{\mathrm{vlm}} \leftarrow \mathcal{M}(\mathbf{x}; \theta_{\mathrm{vlm}}); \hat{\mathbf{y}}_{\mathrm{vlrm}} \leftarrow \mathcal{M}(\mathbf{x}; \theta_{\mathrm{vlrm}})$
   ▷ Compute loss and update parameter.
   $\mathcal{L}_{\mathrm{distill}} \leftarrow \mathrm{KLD}(\hat{\mathbf{y}}_{\mathrm{vlm}}\|\hat{\mathbf{y}}_{\mathrm{vlrm}})$
   $\mathcal{R}_{\mathrm{inject}} \leftarrow \|\mathrm{Sigmoid}(\mathbf{g}) \odot \mathbf{S}\|^2$
   $\mathcal{L} \leftarrow \mathcal{L}_{\mathrm{distill}} - \alpha \cdot \mathcal{R}_{\mathrm{inject}}$
   ▷ Let $\phi = \{g^{(l)}\}_{l=1}^L$ denote all the learnable gates
   $\phi \leftarrow \phi - \eta\nabla_\phi\mathcal{L}$
**end**
**return** $\theta_{\mathrm{vlrm}}$

---

and the pre-trained model weights. Suppose a model $\theta_i$ is finetuned from a pre-trained model $\theta_{pre}$, the task vector is $\tau_i = \theta_i - \theta_{pre}$. When merging $\theta_{1..K}$, the merged model is $\theta_M = \lambda \sum_{i=1}^{K} \tau_i + \theta_{pre}$, where $\lambda$ is the merging coefficient. In our experiments, unless specified, we report the best performance among the merging coefficients among $\lambda \in \{0.1, 0.15, 0.2\}$.

**Ties-Merging** (Yadav et al., 2023) (Trim, Elect Sign & Merge) believes that the conflicts among the task vectors severely affect the performance of the merged model. Ties-Merging solves this problem by eliminating redundant parameters and resolving symbol conflicts. In our experiments, we set the merging coefficient to 0.1 and the pruning density to 0.2.

**DARE** (Yu et al., 2024) (Drop and Rescale) validates the extremely redundant properties of language models. As a pre-processing technique, DARE randomly drops most (90% or even 99%) delta parameters, i.e., task vectors, before merging to potentially mitigate the interference of parameters among models. In our experiments, we set the

merging coefficient to 0.1 and the pruning density to 0.2.

**IP-Merging** (Hu et al., 2025) is a merging method proposed specifically for transferring the math and reasoning capabilities from language reasoning models to multi-modal large language models. Directly merging a strong Math LLM into an VLM fails due to misalignment in parameter spaces and specific reasoning-associated layers. IP-Merging identifies crucial reasoning parameters and projects them into the VLM's subspace to align them, allowing the VLM to directly absorb math reasoning abilities from an off-the-shelf Math LLM without losing its original capabilities. By setting a threshold, IP-Merging selectively merges only the parameter subspaces that exhibit similarity higher than the threshold after projecting them for alignment. Specifically, the higher threshold $T$ usually leads to fewer merged layers. In our experiments, unless specified, we report the best performance among the similarity threshold selections among $T \in \{0.2, 0.3, 0.4\}$.

### C.3. Datasets

**MathVista** (Lu et al., 2024a) is a large-scale multimodal benchmark dataset with thousands of image-based math problems designed to evaluate the visual–mathematical reasoning abilities of VLMs, covering Figure Question Answering (FQA), Geometry Problem Solving (GPS), Math Word Problems (MWP), Textbook Question Answering (TQA), and Visual Question Answering (VQA). In our experiments, we apply the test-mini subset of this dataset for evaluation.

**MathVision** (Wang et al., 2024a) is a benchmark consisting of 3,040 visually grounded math problems from real math competitions across 16 mathematical disciplines and 5 difficulty levels, designed to rigorously evaluate multimodal large models' mathematical reasoning with visual content. In our experiments, we apply the full test set of this dataset for evaluation.

**MathVerse** (Zhang et al., 2024) includes a diverse set of math problems that require understanding and reasoning over both textual and visual information, such as charts, diagrams, and equations. It evaluates whether VLMs truly understand and reason about diagrams in mathematical questions. The testing data can be divided into five domains, i.e., Text Dominant, Text Lite, Vision Intensive, Vision Dominant, and Vision Only. In our experiments, we apply the test-mini subset of this dataset for evaluation.

**MMMU** (Yue et al., 2024) is a large multimodal benchmark combining text and diverse images across six broad disciplines, Art & Design, Business, Science, Health & Medicine, Humanities & Social Science, and Tech & Engineering, to rigorously evaluate the expert-level multimodal understanding and reasoning capabilities of VLMs. In our experiments, we apply the validation set of this dataset for evaluation.

**R1-OneVision** (Yang et al., 2025a) is a large multimodal reasoning dataset with detailed step-by-step annotations across diverse domains, including mathematics, physics, chemistry, biology, and logical deduction, designed to train and evaluate models on complex cross-modal understanding and reasoning tasks. In our experiments, we apply the full dataset for evaluation.

**MMStar** (Chen et al., 2024a) is a manually curated multimodal benchmark consisting of 1,500 vision-essential samples, covering six domains, CP (coarse perception), FP (fine-grained perception), IR (instance reasoning), LR (logical reasoning), ST (science & technology), MA (mathematics), designed to rigorously evaluate the genuine multimodal understanding and reasoning capabilities of vision–language models. In our experiments, we apply the full dataset for evaluation.

**TextVQA** (Singh et al., 2019) is a dataset that evaluates a model's ability to read and reason about text in real-world images. Answering questions in this dataset typically requires extracting Optical Character Recognition (OCR) text from the image and combining it with visual and linguistic context, thereby testing the visual perception capabilities of VLMs. In our experiments, we apply the evaluation subset for evaluation.

**POPE** (Li et al., 2023b) is a benchmark dataset designed to evaluate object hallucination in VLMs by asking simple yes or no questions about the presence of objects in images. In our experiments, we apply the full dataset for evaluation.

**SeedBench** (Li et al., 2023a) is a large multimodal benchmark of human-annotated multiple-choice questions spanning 12 evaluation dimensions, including the comprehension of both image and video modalities. In our experiments, we apply the subset on image modality.

### C.4. Evaluation

For VL reasoning datasets, we apply a step-by-step prompt. Box J.1 shows a case of our prompt. For VL perception datasets, we use the default prompts provided by VLMEvalKit (Duan et al., 2024). During evaluation, all the models follow the greedy decoding strategy. We apply rule-based evaluation metrics for all the tasks.

## D. More Experimental Results

### D.1. More experiments comparing with more models on more benchmarks

We further evaluate the reasoning and perception capabilities of the proposed FRISM. In Tab. A2, based on Qwen2.5VL-7B-Instruct model, we present additional comparisons with

post-training methods, including an efficient post-training method, DRIFT-VL-7B (Huang et al., 2025), and two powerful post-trained models, OpenVLThinker-7B (Deng et al., 2025) and VLAA-Thinker-Qwen2.5VL-7B (Chen et al., 2025a). Additionally, we add experiments on three more visual perception benchmarks:

- **OCRBench** (Liu et al., 2024b), evaluating OCR-related capabilities of multimodal large language models.

- **MME-Perception** (Fu et al., 2025), evaluating perception-level abilities through a diverse set of visual understanding.

- **ChartQA** (Masry et al., 2022), evaluating model's ability to understand chart content.

The results in Tab. A2 show that FRISM is competitive with, and in our setting even slightly outperforms, several stronger post-training methods in average reasoning performance while preserving visual perception substantially better.

In Tab. A3, we compare the performance-cost trade-off of different methods. It can be seen that FRISM achieves impressive performance while requiring significantly lower computational resources. Additionally, FRISM only requires samples from unlabeled visual perception datasets for calibration, while other post-training methods usually require labeled VL reasoning datasets for training.

### D.2. Experimental results on Llama-based models

In our main manuscript, we conduct experiments primarily on the Qwen series models. To further demonstrate the effectiveness of FRISM, we present additional results on Llama-based models in this section.

**Settings.** We merge LLAVA-Next-8B (Li et al., 2024) and DeepThought-8B (Ruliad, 2024), both of which are post-trained from Llama-3-8B (AI@Meta, 2024). Due to that the LLAVA-Next-8B series models can hardly respond following a specified format as instructed, for example, \boxed{}. Therefore, we change the evaluation settings here. We use VLMEvalKit (Duan et al., 2024) for evaluation and apply a step-by-step prompt, using Qwen2.5-32B-Instruct (Team, 2024) as a verifier to validate performance on both VL reasoning and VL perception datasets.

**Results.** The results are presented in Tab. A4. FRISM shows significant improvements in VL reasoning tasks compared to existing methods while still reserving good VL perception capabilities. These results further demonstrate the applicability of FRISM to different kinds of models.

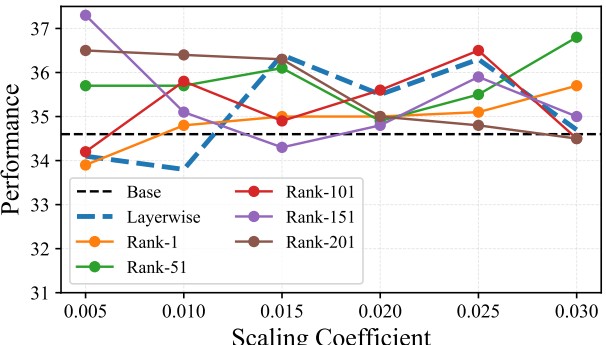

*Figure A1.* Impact of scaling coefficients across subspace ranks when merging Qwen2.5-VL-7B-Instruct and the subspaces of OpenThinker task vector.

### D.3. More comparisons between subspace-level and layer-wise merging

Following the configuration of Fig. 3, we further plot the results by merging the VLMs and the task vectors from another model, OpenThinker3-7B (Guha et al., 2025), in Fig. A1. The results further demonstrate the effectiveness of merging in the subspace level.

## E. Analysis of IP-Merging's Failure Cases

In our experiments, we observed a significant performance degradation when applying the IP-Merging method (Hu et al., 2025) under specific experimental settings, especially when merging Qwen2.5-VL-7B-Instruct and OpenThinker3-7B models under $T = 0.2$ and $T = 0.3$. IP-Merging is designed to transfer reasoning capabilities from LLMs to VLMs by identifying and projecting reasoning-associated parameters. A critical component of this method is the calculation of the merging coefficient, which aims to balance the magnitude of the injected reasoning features with the existing visual features.

IP-Merging calculates the merging coefficient $\lambda$ based on the ratio of their magnitudes to ensure scale alignment:

$$\lambda_n = \frac{\sum_{i=1}^{d} \sigma_{\text{vlm},i}^n}{\sum_{i=1}^{d} \sigma_{\text{lrm},i}^n}, \tag{A2}$$

where $\sigma_i$ is the $i$-th singular value of the task vector. The underlying assumption of IP-Merging is that the magnitude of visual fine-tuning is relatively small compared to reasoning fine-tuning (i.e., $\|\tau_{\text{vis}}\|_2 < \|\tau_{\text{reas}}\|_2$), typically resulting in a coefficient $\lambda < 1$ that serves as a dampening factor to prevent feature disruption. However, this premise does not hold under partial experimental settings. When the visual fine-tuning magnitude is significantly larger than the reasoning fine-tuning magnitude (i.e., $\|\tau_{\text{vis}}\|_2 \gg \|\tau_{\text{reas}}\|_2$),

*Table A2.* Comparison of reasoning and vision performance across more methods on more benchmarks. The base model is Qwen2.5-VL-7B-Instruct. Note that Avg-R and Avg-V are respectively the reasoning and vision scores computed by first normalizing each benchmark score by the corresponding base VLM score and then averaging across benchmarks.

| | MVista | MVerse | MVision | MMMU | R1-OV | MMStar | **Avg-R** | TextVQA | POPE | SeedBench | MME | OCRBench | ChartQA | **Avg-V** |
|---|---|---|---|---|---|---|---|---|---|---|---|---|---|---|
| Base | 68.1 | 41.2 | 25.6 | 55.1 | 34.6 | 60.1 | 100% | 85.5 | 86.4 | 77.0 | 1697.9 | 87.7 | 86.2 | 100% |
| *Post-Training Methods* | | | | | | | | | | | | | | |
| DRIFT | 70.4 | 41.8 | 25.7 | 55.7 | 34.9 | 59.7 | 101.3% | 84.1 | 84.3 | 70.7 | 1510.7 | 87.6 | 80.2 | 94.9% |
| OpenVLThinker-7B | 69.4 | 41.5 | 26.5 | 54.1 | 35.0 | 62.5 | 101.7% | 84.2 | 82.6 | 74.6 | 1411.6 | 83.4 | 84.2 | 94.5% |
| VLAA-Thinker | 71.3 | 42.8 | 26.8 | 57.8 | 35.4 | 61.3 | 103.8% | 82.9 | 86.2 | 75.2 | 1700.3 | 86.9 | 84.3 | 98.6% |
| *Merging Methods* | | | | | | | | | | | | | | |
| IP-Merging | 67.2 | 40.8 | 25.5 | 55.9 | 36.0 | 60.7 | 100.6% | 82.9 | 87.1 | 76.9 | 1694.8 | 88.4 | 81.6 | 98.8% |
| TaskArithmetic | 65.4 | 37.8 | 24.2 | 53.7 | 36.6 | 60.7 | 97.9% | 83.9 | 86.8 | 76.5 | 1674.9 | 88.5 | 72.9 | 97.0% |
| **FRISM (Ours)** | 70.3 | 41.8 | 26.4 | 57.8 | 37.5 | 62.4 | **104.2%** | 85.0 | 87.1 | 76.9 | 1678.9 | 87.9 | 84.6 | **99.6%** |

*Table A3.* Comparison of training efficiency and performance.

| Method | #GPUs | Training Time | Avg-Reason | Avg-Percep |
|---|---|---|---|---|
| Base | – | – | 100% | 100% |
| IP-Merging | – | – | 100.6% | 98.8% |
| DRIFT | 8 | ∼2 hours | 101.3% | 94.9% |
| OpenVLThinker-7B | 8 | >1 day | 101.7% | 94.5% |
| VLAA-Thinker-7B | 8 | ∼6 hours | 103.8% | 98.6% |
| **FRISM (Ours)** | 1 | **30 mins** | **104.2%** | **99.6%** |

the coefficient $\lambda$ becomes erroneously large, resulting in a failure in performance.

# F. Theoretical Analysis

## F.1. Subspace-level filtering mechanism

We provide a theoretical analysis of the proposed FRISM framework. We formulate the reasoning injection process as a constrained optimization problem within the parameter space and demonstrate that our subspace-level gating mechanism serves as an adaptive spectral filter that maximizes reasoning gain, bounded by visual degradation.

Let $\theta_{\text{vlm}}$ denote the parameters of the original VLM. The visual capability is measured by the loss function $\mathcal{L}_{\text{vis}}(\theta)$. We assume the pre-trained VLM has converged to a local minimum, implying the gradient $\nabla \mathcal{L}_{\text{vis}}(\theta_{\text{vlm}}) \approx 0$. Consider an additive parameter update $\Delta\theta$ derived from the reasoning model. The visual loss of the merged model can be approximated via Taylor expansion:

$$
\begin{aligned}
&\mathcal{L}_{\text{vis}}(\theta_{\text{vlm}} + \Delta\theta) \\
&\approx \mathcal{L}_{\text{vis}}(\theta_{\text{vlm}}) + \nabla \mathcal{L}_{\text{vis}}(\theta_{\text{vlm}})^\top \Delta\theta + \frac{1}{2}\Delta\theta^\top \mathbf{H}\Delta\theta \\
&\approx \frac{1}{2}\Delta\theta^\top \mathbf{H}\Delta\theta,
\end{aligned} \quad \text{(A3)}
$$

where $\mathbf{H} = \nabla^2 \mathcal{L}_{\text{vis}}(\theta_{\text{vlm}})$ is the Hessian matrix. The zeroth and first-order terms are neglected due to local optimality. Let:

$$
\delta_{\text{vis}}(\Delta\theta) = \frac{1}{2}\Delta\theta^\top \mathbf{H}\Delta\theta \quad \text{(A4)}
$$

denote the visual degradation caused by merging.

Existing merging methods apply a scalar coefficient $\lambda$ to the task vector $\tau_{\text{lrm}}$, yielding $\Delta\theta = \lambda \cdot \tau_{\text{lrm}}$. The visual degradation is $\delta_{\text{vis}} = \lambda^2 \tau_{\text{lrm}}^\top \mathbf{H} \tau_{\text{lrm}}$. If $\tau_{\text{lrm}}$ overlaps with the eigenvectors of $\mathbf{H}$ corresponding to large eigenvalues (high curvature directions), resulting in significant visual degradation.

In our FRISM, we decompose the update into SVD subspaces:

$$
\Delta\theta = \sum_{i=1}^{N} \lambda_i B_i, \quad \text{(A5)}
$$

where $B_i$ represents the $i$-th component (scaled by singular value) and $\lambda_i = \lambda_{\text{lrm}} \cdot \text{Sigmoid}(\mathbf{g}^{(l)})$. We formulate the total objective as maximizing reasoning intensity (proxied by the norm of the merged task vector) while minimizing visual degradation:

$$
\mathcal{L} = \mathcal{L}_{\text{vis}} - \alpha \mathcal{L}_{\text{reason}} = \frac{1}{2}\Delta\theta^\top \mathbf{H}\Delta\theta - \alpha\|\Delta\theta\|_F^2. \quad \text{(A6)}
$$

Substituting the subspace decomposition into the loss:

$$
\mathcal{L} = \frac{1}{2}\sum_{i,j}\lambda_i\lambda_j \text{Tr}(B_i^\top \mathbf{H} B_j) - \alpha \sum_{i,j}\lambda_i\lambda_j \text{Tr}(B_i^\top B_j). \quad \text{(A7)}
$$

Since $\{B_i\}$ form an orthogonal basis in the parameter space, we have:

$$
\text{Tr}(B_i^\top B_j) = \|B_i\|_F^2 \delta_{ij}. \quad \text{(A8)}
$$

Furthermore, we adopt the diagonal approximation for the Hessian interaction terms, assuming different SVD subspaces are approximately decoupled with respect to the visual loss curvature (i.e., $\text{Tr}(B_i^\top \mathbf{H} B_j) \approx 0$ for $i \neq j$). This simplifies the objective to a decoupled sum:

$$
\mathcal{L} \approx \sum_{i=1}^{N}\left(\frac{1}{2}\lambda_i^2 h_i - \alpha\lambda_i^2\|B_i\|_F^2\right), \quad \text{(A9)}
$$

where $h_i = \text{Tr}(B_i^\top \mathbf{H} B_i)$ represents the projected visual curvature along the $i$-th subspace.

*Table A4.* Experiments of merging LLAVA-Next-8B with DeepThought-8B model.

| Methods | VL Reasoning Benchmarks | | | | VL Perception Benchmarks | | |
|---|---|---|---|---|---|---|---|
| | MVista | MVerse | MVision | Avg | TextVQA | POPE | Avg |
| Base | 29.0 | 10.7 | 9.5 | 16.4 | 61.2 | 87.1 | 74.2 |
| TA | 31.4 | 11.4 | 10.9 | 17.9 | 61.2 | **86.7** | **74.0** |
| IP-Merging | 29.8 | 9.8 | 9.9 | 16.5 | 61.3 | 86.2 | 73.8 |
| FRISM (Ours) | **32.1** | **12.8** | **10.9** | **18.6** | 61.6 | 86.3 | **74.0** |

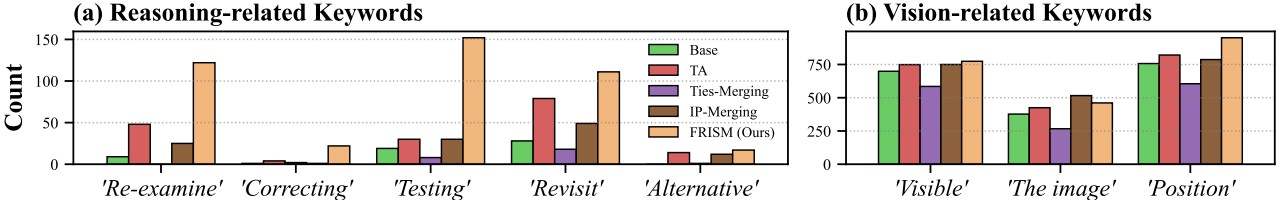

*Figure A2.* Keyword Analysis of different merging methods for (a) reasoning-related keywords and (b) vision-related keywords.

Analyzing the gradient of Eq. A9 with respect to the gate $\lambda_i$:

$$\frac{\partial \mathcal{L}}{\partial \lambda_i} = \left( h_i - 2\alpha \|B_i\|_F^2 \right) \cdot \lambda_i. \tag{A10}$$

This derivative theoretically proves the spectral filtering mechanism of FRISM. The optimization direction is determined by the sign of the bracketed term, creating two distinct regimes:

**1. Suppression Regime** ($h_i > 2\alpha \|B_i\|_F^2$)**:** If the rank-$i$ subspace $B_i$ aligns with high-curvature directions of the visual Hessian (large $h_i$), the visual degradation cost outweighs the reasoning gain. The gradient is positive, driving $\lambda_i \to 0$. This guarantees the preservation of visual capabilities by filtering out conflicting components.

**2. Injection Regime** ($h_i < 2\alpha \|B_i\|_F^2$)**:** If the rank-$i$ subspace $B_i$ lies in the flat directions of the visual landscape (small $h_i$) or represents a significant reasoning component (large $\|B_i\|_F^2$), the gradient is negative, driving $\lambda_i$ to converge to a higher value.

Therefore, FRISM acts as an optimal filter that selectively injects reasoning capabilities based on the local curvature of the visual loss landscape.

### F.2. Loss term analysis

In this section, we provide a formal justification for using the maximization of the effective task vector norm, which is defined by:

$$\mathcal{L}_{\text{inject}} = -\sum_{l=1}^{L} \|\mathbf{S}_{\text{eff}}^{(l)}\|^2 = -\sum_{l=1}^{L} \|\text{Sigmoid}(\mathbf{g}^{(l)}) \odot \mathbf{S}^{(l)}\|^2, \tag{A11}$$

as a valid mathematical proxy for injecting reasoning capabilities in the absence of labeled reasoning supervision.

We analyze the validity of $\mathcal{L}_{inject}$ considering the full merged parameter formulation which includes the visual task vector $\tau_{vlm}$.

Define the merged model parameters:

$$\theta_{\text{merged}} = \theta_{\text{base}} + \tau_{\text{vlm}} + \Delta\theta_{\text{lrm}}^{\text{eff}} \tag{A12}$$

We measure the proximity of this model to the LRM. The squared Euclidean distance is:

$$\begin{aligned} J(\lambda) &= \|\theta_{\text{lrm}} - \theta_{\text{merged}}\|_F^2 \\ &= \|\tau_{\text{lrm}} - (\tau_{\text{vlm}} + \Delta\theta_{\text{lrm}}^{\text{eff}})\|_F^2 \end{aligned} \tag{A13}$$

The reasoning residual is $\Delta_{\text{res}} = \tau_{\text{lrm}} - \tau_{\text{lrm}}^{\text{eff}} = \sum(1 - \lambda_i)B_i$. Expand the norm and we have:

$$\begin{aligned} J(\lambda) &= \|\Delta_{\text{res}} - \tau_{\text{vlm}}\|_F^2 \\ &= \|\Delta_{\text{res}}\|_F^2 + \|\tau_{\text{vlm}}\|_F^2 - 2\langle\Delta_{\text{res}}, \tau_{\text{vlm}}\rangle \end{aligned} \tag{A14}$$

A core premise of FRISM is that reasoning capabilities and visual representations reside in distinct subspaces. Mathematically, this implies approximate orthogonality between the reasoning basis $B_i$ and the visual task vector $\tau_{vlm}$:

$$\begin{aligned} \langle B_i, \tau_{\text{vlm}}\rangle &\approx 0, \\ \langle\Delta_{\text{res}}, \tau_{\text{vlm}}\rangle &\approx 0. \end{aligned} \tag{A15}$$

Under this hypothesis, the cross-term vanishes. Therefore, the distance function simplifies to:

$$J(\lambda) \approx \sum_{i=1}^{N} (1 - \lambda_i)^2 \|B_i\|_F^2 + C \tag{A16}$$

## Model Inputs:

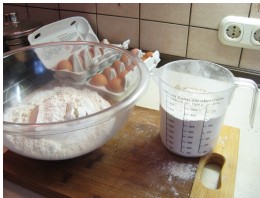

Hint : Please answer the question requiring an integer answer and provide the final value , e .g ., 1 , 2 , 3 , at the end .↵ Question : what is the total volume of the measuring cup ? ( Unit : g )↵↵ Please reason step by step , and put your final answer within \ boxed {}.

## Base VLM

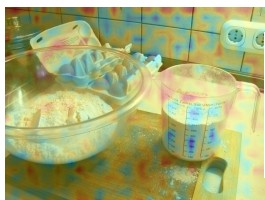

Hint : Please answer the question requiring an integer answer and provide the final value , e .g ., 1 , 2 , 3 , at the end .↵ Question : what is the total volume of the measuring cup ? ( Unit : g )↵↵ Please reason step by step , and put your final answer within \boxed {}.

## Task Arithmetic

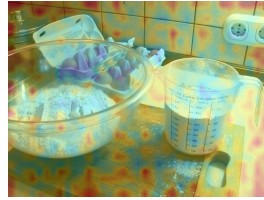

Hint : Please answer the question requiring an integer answer and provide the final value , e .g ., 1 , 2 , 3 , at the end .↵ Question : what is the total volume of the measuring cup ? ( Unit : g )↵↵ Please reason step by step , and put your final answer within \ boxed {}.

## IP-Merging

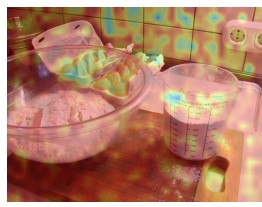

Hint : Please answer the question requiring an integer answer and provide the final value , e .g ., 1 , 2 , 3 , at the end .↵ Question : what is the total volume of the measuring cup ? ( Unit : g )↵↵ Please reason step by step , and put your final answer within \boxed {}.

## FRISM (Ours)

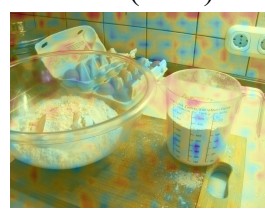

Hint : Please answer the question requiring an integer answer and provide the final value , e .g ., 1 , 2 , 3 , at the end .↵ Question : what is the total volume of the measuring cup ? ( Unit : g )↵↵ Please reason step by step , and put your final answer within \ boxed {}.

*Figure A3.* Qualitative comparison of attention maps across different model merging methods on a sample from MathVista dataset.

where $C$ is a constant independent of the gating parameters.

Note that $0 < \lambda_i < 1$, therefore, minimizing the distance to the LRM is equivalent to minimizing $\sum(1 - \lambda_i)^2 \|B_i\|_F^2$. Thus, maximizing the injection loss $\mathcal{L}_{\text{inject}}$ is a strategy to minimize the parameter distance to the LRM, thereby minimizing the theoretical upper bound of the reasoning performance gap, even in the presence of the visual task vector

$\tau_{\text{vlm}}$.

## G. Visualization Results

In Fig. A3, we present a qualitative comparison of attention maps across different methods on a sample from the MathVista (Lu et al., 2024a) dataset. We visualize the regions focused on by the models in the image. As observed, Base

VLM focuses on the markings of the measuring cup. The models merged by the baseline methods usually exhibit a shift in the visual attention region. In contrast, our FRISM demonstrates visual attention regions similar to those of the base model. Furthermore, in terms of text attention regions, baseline methods tend to exhibit attention dispersion and focus on the format instrctions or syntactic artifacts, ignoring the critical chain-of-thought triggers. In contrast, FRISM maintains high attention weights on these key reasoning keywords, including "reason" and "step". This demonstrates that our method achieves effective reasoning capability injection without compromising visual robustness.

## H. FRISM Technique Details

### H.1. Implementation details

In Sec. 3.1.2, we mentioned that for practical using, the reasoning injection loss $\mathcal{L}_{\text{inject}}$ is normalized before training, as follows:

Before training:

$$\mathcal{L}_{\text{inject}}^{\text{init}} = -\sum_{l=1}^{L} \|\text{Sigmoid}(\mathbf{g}_{\text{init}}{}^{(l)}) \odot \mathbf{S}^{(l)}\|^2. \quad (A17)$$

Practically, as we mentioned in Sec. 3.1.2, $\mathbf{g}^{(l)}$ are initalized by all-zero values. Therefore,

$$\mathcal{L}_{\text{inject}}^{\text{init}} = -\sum_{l=1}^{L} \|\frac{1}{2}\mathbf{S}^{(l)}\|^2. \quad (A18)$$

During Training, the $\mathcal{L}_{\text{inject}}$ are normalized by the absolute value of their initial values:

$$\mathcal{L}_{\text{inject}}^{\text{training}} = -\frac{\sum_{l=1}^{L} \|\text{Sigmoid}(\mathbf{g}^{(l)}) \odot \mathbf{S}^{(l)}\|^2}{\mathcal{L}_{\text{inject}}^{\text{init}}}. \quad (A19)$$

### H.2. Hyper-parameter analysis

Next we analyze the two critical hyper-parameters in FRISM: the injection coefficient $\alpha$ and the global merging coefficient $\lambda_{\text{lrm}}$. Their interaction determines the trade-off between reasoning injection and visual preservation.

**Injection Coefficient** $\alpha$ balances the dual objectives of visual preservation ($\mathcal{L}_{\text{distill}}$) and reasoning maximization ($\mathcal{L}_{\text{inject}}$). Our theoretical analysis in Appendix F, Eq A10 reveals its physical significance as a threshold controller for the subspace filter. $\frac{\partial \mathcal{L}}{\partial \lambda_i} = (h_i - 2\alpha\|B_i\|_F^2) \cdot \lambda_i$.

When $\alpha$ is small, the visual degradation term, $h_i$, dominates, causing the model to suppress most subspaces to protect visual capabilities. This leads to high visual performance but insufficient reasoning gain.

*Table A5.* Count of Keywords generated by different methods when merging Qwen2.5-VL-32B with QwQ-32B models.

| Keyword | Reasoning-related | | | | | | | | Vision-related | | |
|---|---|---|---|---|---|---|---|---|---|---|---|
| | "Re-examine" | "Correcting" | "Misinterpret" | "However" | "Incorrect" | "Alternative" | "Testing" | "Revisit" | "The image" | "Visible" | "Position" |
| Base | 9 | 1 | 16 | 1225 | 158 | 0 | 19 | 28 | 377 | 699 | 757 |
| TA | 48 | 4 | 40 | 1690 | 214 | 14 | 30 | 79 | 425 | 749 | 821 |
| Ties | 0 | 2 | 63 | 828 | 47 | 1 | 8 | 18 | 267 | 585 | 605 |
| IP | 25 | 1 | 18 | 1042 | 282 | 12 | 30 | 49 | 516 | 750 | 787 |
| FRISM | 122 | 22 | 47 | 2097 | 325 | 17 | 152 | 111 | 461 | 774 | 951 |

As $\alpha$ increases, the reasoning gain term, $2\alpha\|B_i\|_F^2$, counteracts the visual degradation term. This lowers the filtering bar, allowing subspaces with higher potential visual conflict (larger curvature in $H$) to be injected. While this maximizes reasoning capabilities, excessive $\alpha$ may degrade visual perception by overriding the distillation constraint.

Empirically, we recommend selecting $\alpha$ so that the magnitude of $\mathcal{L}_{\text{inject}}$ is comparable to $\mathcal{L}_{\text{distill}}$ during the initial training phase to ensure balanced gradient updates. The value of $\alpha$ is usually $\in [0.1, 0.3]$ for our experiments.

**Merging Coefficient** $\lambda_{\text{lrm}}$ is the injection capacity bound in our FRISM. The global coefficient $\lambda_{lrm}$ acts as a scaling factor for the whole task vector. Since the learnable gate passes through a Sigmoid function, i.e., $\text{Sigmoid}(\mathbf{g}^{(l)}) \in (0, 1)$, $\lambda_{lrm}$ essentially defines the upper bound of the injection magnitude for any given subspace.

If $\lambda_{\text{lrm}}$ is set too low, even with $g \to \infty$, the effective singular value $S_{\text{eff}}$ may remain too small to influence the model's behavior, leading to under-fitting of the reasoning task. Conversely, an excessively high $\lambda_{\text{lrm}}$ can lead to unstable initialization before the gates adapt. In our experiments, the global merging coefficient $\lambda_{\text{lrm}}$ is searched in $\{0.1, 0.15, 0.2\}$. which is adjustable depending on the magnitude of the LRM task vector $\tau_{\text{lrm}}$.

## I. More results for keyword analysis

In Tab.A5, we show more keyword analysis results on Math-Vision when merging Qwen2.5-VL-32B-Instruct and QwQ-32B. Our FRISM shows both strong self-reflection and visual focus capabilities. A visualized comparison is shown in Fig. A2.

## J. Case Study

We present a case study from the MathVision (Wang et al., 2024a) dataset. The image in the question, prompt for the model, and the correct answer are shown in Box J.1. In this section, we compare the responses generated by the base

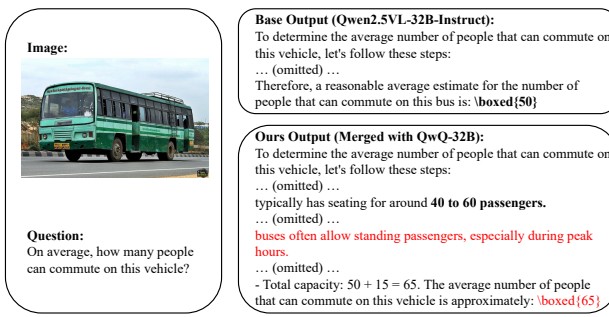

*Figure A4.* A failure case of the proposed FRISM. The merged model first compute the correct answer before changing it to the wrong one due to overthinking.

VLM Qwen2.5-VL-7B-Instruct and the models merged with DeepSeek-R1-Distill-Qwen-7B through Task Arithmetic, IP-Merging, and ours. The generated responses are respectively shown in Box J.2, J.3, J.4, J.5. It can be seen that the response generated by ours shows reasoning and re-thinking capabilities and finally outputs the correct answer.

## K. Failure Cases of FRISM

In Fig. A4, we present a failure case of the proposed FRISM from the MathVista (Lu et al., 2024a) dataset. We compare the responses generated by the base VLM Qwen2.5-VL-32B-Instruct and the models merged with QwQ-32B through our FRISM. It can be seen that though showing significant performance improvements, merging VLMs with LRMs may introduce overthinking issues, which can lead to some failure cases.

## L. Limitations and Future Works

Despite the convincing results, the proposed FRISM suffers from the following limitations.

- After merging with LRMs, the output length of VLMs can significantly increase, thus increasing the inference time and computation costs, which is a common problem within this field. Future studies may include jointly optimizing performance and efficiency.

- Our reasoning objective uses injected subspace magnitude as a proxy because labeled VL reasoning data are unavailable in our setting. This choice is effective empirically but it is still a proxy rather than a direct supervision signal for reasoning transfer. Future studies may focus on the supervision signal for reasoning transfer during label-free training.

- FRISM is fine-grained subspace-level merging based on posterior findings. The theoretical principles un-

derlying this are worth further analysis. Also, as a common limitation of task vector-based merging methods, FRISM cannot be generalized to models with different sizes or structures.

Further transferring reasoning capabilities from reasoning language models to models with more modalities, model sizes, or even different structures is a significant future direction.

---

### Box J.1: Question and the correct answer

**Image**:

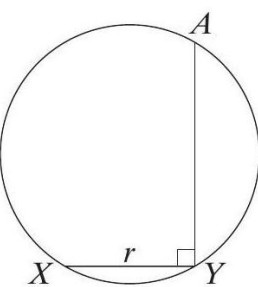

**Prompt**:
Hint: Please answer the question requiring an answer.
Question: On the circumference of radius $r$ three points $X, Y$ and $A$ are marked such that $XY = r$, $XY \perp AY$ (see the figure). How many degrees has the angle $XAY$?
Please reason step by step, and put your final answer within \boxed{}.
**Correct Answer**:
30°

---

### Box J.2: Response generated by base VLM (Qwen2.5-VL-7B-Instruct).

To solve for the measure of angle $\angle XAY$, we analyze the given geometric configuration.
1. **Identify the given information:**

   - The circle has radius $r$.
   - Points $X$ and $Y$ lie on the circumference of the circle.
   - Point $A$ also lies on the circumference of the circle.
   - $XY = r$.
   - $XY \perp AY$.

2. **Analyze the right triangle:**

   Since $XY \perp AY$, triangle $AXY$ is a right triangle with

   $$\angle AXY = 90°.$$

3. **Determine the relationship between the sides:**

   In a right triangle, the hypotenuse is the longest side. Here, $AY$ is the hypotenuse since it connects two points on the circumference and is perpendicular to $XY$.

4. **Use the properties of the circle:**

   Because $X$ and $Y$ lie on the circumference and $XY = r$, the segment $XY$ is a chord of the circle. Given that $XY \perp AY$, point $A$ lies directly above or below the midpoint of $XY$ along a diameter of the circle.

5. **Calculate the angle:**

   In a right triangle inscribed in a semicircle, the angle subtended by the diameter at any point on the circumference is always 90°. Therefore,

   $$\angle XAY = 90°.$$

   $$\boxed{90°}$$

---

**Box J.3: Response generated by Task Arithmetic.**

---

To solve for the angle $\angle XAY$ in the given problem, we analyze the geometric configuration and apply properties of circles and right triangles.

1. **Identify the given information:**

   - The circle has radius $r$.
   - Points $X$, $Y$, and $A$ lie on the circumference of the circle.
   - $XY = r$.
   - $XY \perp AY$.

2. **Analyze triangle $\triangle AXY$:**

   Since $XY$ is a chord of the circle and $XY \perp AY$, segment $AY$ must be a radius of the circle because the perpendicular from the center of a circle to a chord bisects the chord. Therefore,

   $$AY = r.$$

3. **Determine the type of triangle:**

   Triangle $\triangle AXY$ is a right triangle with
   $$\angle AXY = 90°$$
   because $XY \perp AY$. Since both $AX$ and $AY$ are radii of the circle, we have

   $$AX = AY = r.$$

4. **Calculate the angles in the triangle:**

   Triangle $\triangle AXY$ is isosceles, so the base angles $\angle XAY$ and $\angle AYX$ are equal. Let

   $$\angle XAY = \theta.$$

   Then,
   $$\angle AYX = \theta.$$

5. **Use the angle sum of a triangle:**

   The sum of the angles in a triangle is $180°$. Hence,

   $$\theta + \theta + 90° = 180°.$$

   Simplifying,
   $$2\theta + 90° = 180°,$$

   which gives
   $$2\theta = 90° \quad \Rightarrow \quad \theta = 45°.$$

Thus, the angle $\angle XAY$ is

$$\boxed{45°}.$$

**Box J.4: Response generated by the model obtained through IP-Merging.**

To solve for the measure of angle $\angle XAY$, we analyze the given geometric configuration.

1. **Identify the given information:**

   - The circle has a radius $r$.
   - Points $X$ and $Y$ lie on the circumference of the circle.
   - Point $A$ is also on the circumference of the circle.
   - $XY = r$.
   - $XY \perp AY$.

2. **Analyze the right triangle:** Since $XY \perp AY$, triangle $AXY$ is a right triangle with

$$\angle AXY = 90°.$$

3. **Determine the relationship between the sides:** In a right triangle, the hypotenuse is the longest side. Here, $AY$ is the hypotenuse because it connects two points on the circumference of the circle and is perpendicular to $XY$.

4. **Use the properties of the circle:** Since $X$ and $Y$ are points on the circumference and $XY = r$, the segment $XY$ is a chord of the circle. Given that $XY \perp AY$, point $A$ must lie directly above or below the midpoint of $XY$ along a diameter of the circle.

5. **Calculate the angle:** In a right triangle inscribed in a semicircle, the angle subtended by the diameter at any point on the circumference is always $90°$. Therefore,

$$\angle XAY = 90°.$$

Thus, the measure of angle $\angle XAY$ is

$$\boxed{90°}.$$

---

**Box J.5: Response generated by the model obtained through FRISM.**

To solve the problem, we analyze the given geometric configuration using properties of circles and right triangles.

1. **Identify the given information:**

   - The circle has radius $r$.
   - Points $X$, $Y$, and $A$ lie on the circumference of the circle.
   - $XY = r$.
   - $XY \perp AY$.

2. **Analyze triangle $\triangle XYO$:**

   Since $XY = r$ and $XO = YO = r$ (both are radii of the circle), triangle $\triangle XYO$ is equilateral.

3. **Determine the central angle subtended by arc $XY$:**

   In an equilateral triangle inscribed in a circle, each side subtends a central angle of $60°$. Therefore,

   $$\angle XOY = 60°.$$

4. **Analyze the right triangle $\triangle AXY$:**

   Since $XY \perp AY$, triangle $\triangle AXY$ is a right triangle with

   $$\angle AYX = 90°.$$

   Let

   $$\angle XAY = \theta.$$

5. **Use the angle sum of a triangle:**

   The three angles of $\triangle AXY$ are

   $$\angle AYX = 90°, \quad \angle XAY = \theta, \quad \angle AXY = 90° - \theta.$$

   Their sum is

   $$90° + \theta + (90° - \theta) = 180°,$$

   which confirms the setup is consistent.

6. **Determine $\angle XAY$:**

   Since $A$ lies on the circumference, the inscribed angle $\angle XAY$ subtends the same arc $XY$ as the central angle $\angle XOY$. An inscribed angle equals half the corresponding central angle, so

   $$\angle XAY = \frac{1}{2}\angle XOY = \frac{60°}{2} = 30°.$$

   $$\boxed{\angle XAY = 30°}$$

