# OpenReview forum: "FRISM: Fine-Grained Reasoning Injection via Subspace-Level Model Merging for Vision–Language Models"
_ICML.cc/2026/Conference — ICML 2026 regular_

### Official Review · Reviewer_dMyT · 2026-03-06

**Soundness:** 3
**Presentation:** 3
**Significance:** 3
**Originality:** 3
**Overall Recommendation:** 4
**Confidence:** 4

**Summary:**

This paper proposes FRISM, a subspace-level model merging method for transferring reasoning ability from large reasoning models to vision-language models. The key idea is to decompose reasoning task vectors into finer-grained subspaces and learn subspace-wise merging coefficients, while using label-free self-distillation to preserve visual capability. The experiments show improved reasoning performance with relatively limited degradation on perception tasks.

**Compliance With Llm Reviewing Policy:**

Affirmed.

**Final Justification:**

My concerns have been adequately addressed.

**Key Questions For Authors:**

1. The paper argues that reasoning capabilities can be injected through particular subspaces. Is there any stronger evidence that these subspaces are genuinely reasoning-related, rather than simply directions that work well under the current training setup?

2. Can the authors do more to separate the effect of the subspace-level merging from the effect of the self-distillation objective? Right now the two are somewhat entangled.

3. How robust is FRISM when the VLM and LRM are less aligned, or come from more heterogeneous model families?

4. Could the authors expand the discussion of limitations a bit more? In particular, I think it would be helpful to comment on assumptions behind the subspace decomposition and on cases where the merging may hurt visual capability or fail to transfer reasoning cleanly.

**Limitations:**

yes

**Strengths And Weaknesses:**

Strengths

1. The paper addresses a relevant problem. Efficiently improving reasoning in VLMs without expensive multimodal post-training is a worthwhile direction.

2. The main idea is clean and easy to follow. Moving from layer-wise merging to subspace-level merging feels like a natural extension, and the method is reasonably well motivated.

3. The empirical results are generally good. FRISM improves reasoning performance while keeping perception performance relatively stable in most of the reported settings.

4. The method is also practically appealing, since it is much lighter than full post-training.

Weaknesses

1. The main intuition of the paper — that reasoning can be injected through distinct subspaces — is interesting, but at the moment it is still supported more by empirical results than by a truly convincing explanation.

2. It is also not completely clear how much of the gain should be attributed to the subspace-level merging itself, and how much comes from the self-distillation / dual-objective training.

3. The experiments are promising, but I would still like to see broader evidence, especially on more heterogeneous VLM/LRM pairs or more diverse downstream settings.

4. The paper does not really discuss limitations in much depth. The impact statement is very brief, and I think the paper would benefit from a more explicit discussion of where FRISM may fail or where its assumptions may not hold.

---

> ### Author Rebuttal · Authors · 2026-03-31
>
> We thank the reviewer for the constructive comments and for recognizing the strengths of our work, including its clear motivation, clean technical idea, and practical potential.
> We respond to the concerns as follows:
> ## Weakness1:Method explanation
> Our claim is that some subspaces in the LRM task vector are more reasoning-relevant and transfer-useful than others for improving VL reasoning while preserving visual capability, leading to better performance than layer-level methods. Our evidence is functional rather than definitive: under the current model pair and training setup, some SVD subspaces are more effective for reasoning transfer, not necessarily intrinsically reasoning-only.
> We apply a learning-based method for identifying beneficial subspaces because it is difficult to identify a suitable criterion for our task. Empirically, different subspaces exhibit different optimal scaling coefficients, as shown in Fig. 3. The dual-objective acts as a filter against visually harmful directions, and removing SVD weakens performance in Table 4. These findings are consistent with prior work showing compact encoding of high-level/reasoning-related functions [R1] and the usefulness of partial subspaces for merging [R2]. We will revise the wording in revision accordingly.
> ## Weakness2 & Question2:Significance of subspace-level merging
> Our current ablations in Tab. 4 already provide a first decomposition: removing SVD reduces the average reasoning score from 49.4 to 47.9, while removing the injection objective reduces it to 47.8. This suggests that both components matter and are complementary, rather than the gains being explained solely by self-distillation or solely by subspace decomposition.
> Furthermore, we present some results of the performance of Qwen2.5-VL-7B-Instruct on GSM8K after merging different subspaces of an LRM. The performance of base VLM is 68.6. It can be seen that subspaces of different ranks can bring about improvements, which is positively related to the scaling factor. This further suggests that post-training updates are not functionally homogeneous across subspaces, which is consistent with our motivation for subspace-level merging.
> |rank\alpha|0.01|0.02|0.03|0.04|
> |-|-|-|-|-|
> |1|71.1|71.9|72.3|73.2|
> |101|69.1|69.7|70.1|71.2|
> |201|69.2|69.5|69.4|69.7|
> ## Weakness3 & Question3:More experiments
> In our paper, FRISM has been evaluated on multiple VLM/LRM pairs beyond the main Qwen2.5-VL and DeepSeek setting, including ThinkLite-VL, OpenThinker3, and InternVL3, where FRISM remains consistently competitive or best.
> Notably, even the main setting of merging Qwen2.5-VL-7B-Instruct and DeepSeek-R1-Distill-Qwen models is challenging because the LRM update magnitude is substantially larger than that of the VLM, which makes direct merging unstable for existing methods. FRISM nonetheless remains consistently beneficial.
> To further demonstrate the robustness of FRISM, we conduct additional experiments. We merge Qwen2.5-VL-7b-Instruct and DLER-R1-7B-Research, a language reasoning model further post-trained on DeepSeek-Distill-Qwen-7B, making the experimental setting more difficult. The results are as follows:
> ||MVista|R1-OV|MMStar|TextVQA|OCRBench|
> |-|-|-|-|-|-|
> |Base|68.1|34.6|60.1|85.5|87.7|
> |TA|68.0|36.1|61.6|83.7|87.6|
> |IP|68.5|35.2|60.6|83.1|87.4|
> |**FRISM(Ours)**|69.3|37.3|62.1|85.3|87.6|
>
> We will include more detailed results in revision. Moreover, we also include more benchmarks and comparisons with some post-training methods. Please kindly check our response to Reviewer 8U3c for details. We will add these results in revision.
> Though showing impressive performance, like other merging methods, FRISM is based on pretrain-finetune paradigm. Transferring across different pretrained families remains an important direction for our future work.
>
> ## Weakness4 & Question4:Limitations
> Here we discuss more limitations:
> - After merging with LRMs, the output length of VLMs can significantly increase, thus increasing the inference time and computation costs, which is a common problem within this field. Future studies may include jointly optimizing performance and efficiency.
> - Our reasoning objective uses injected subspace magnitude as a proxy because labeled VL reasoning data are unavailable in our setting. This choice is effective empirically but it is still a proxy rather than a direct supervision signal for reasoning transfer. Future studies may focus on the supervision signal for reasoning transfer during label-free training.
> - FRISM is fine-grained subspace-level merging based on posterior findings. The theoretical principles underlying this are worth further analysis.
> We will revise the paper accordingly. We will also add some failure cases and corresponding analysis in revision.
> ## Refs
> [R1]Lssf: Safety alignment for large language models through low-rank safety subspace fusion. ACL 2025.
>
> [R2]Delta-come: Training-free delta-compression with mixed-precision for large language models. NeurIPS 2024.

---

> > ### Author Rebuttal · Reviewer_dMyT · 2026-04-03
> >
> > Thank you to the authors for the detailed rebuttal. My previous questions have been resolved. I will keep my initial positive score.

---

> > > ### Author Response · Authors · 2026-04-03
> > >
> > > We sincerely thank you for the positive feedback. We appreciate your recognition of our work. Your constructive review has made a significant contribution to improving the quality of our paper. We will further improve the final version of our paper based on your comments. Thank you again for your time and support.

---

### Official Review · Reviewer_5im7 · 2026-03-10

**Soundness:** 2
**Presentation:** 3
**Significance:** 2
**Originality:** 3
**Overall Recommendation:** 3
**Confidence:** 4

**Summary:**

This paper studies how to transfer reasoning ability from large reasoning models into vision-language models through model merging, while avoiding the visual capability degradation commonly observed in prior merging methods. The main proposal, FRISM, replaces coarse layer-level merging with subspace-level merging: it decomposes the LRM task vector via SVD, freezes the singular bases, and learns lightweight per-subspace gates that modulate the effective singular values before injecting them into the VLM. To avoid requiring scarce multimodal reasoning annotations, the paper introduces a label-free self-distillation scheme that uses perception data only, combining a visual-preservation objective with an injection-strength objective. Experiments across multiple Qwen2.5-VL scales and several additional model pairs show that FRISM usually improves average reasoning performance over standard merging baselines while largely preserving performance on a small set of perception benchmarks.

**Compliance With Llm Reviewing Policy:**

Affirmed.

**Ethical Review Concerns:**

Reviewer Console
https://openreview.net/group?id=ICML.cc/2026/Conference/Reviewers#assigned-submissions

**Final Justification:**

After reading the rebuttal and discussion, my main concerns are only partially resolved. Much of the new evidence is still deferred to the revision, the mechanistic claim remains suggestive rather than fully established, and the practical value of post-hoc model merging versus continued multimodal training is still not entirely convincing to me. Overall, the rebuttal improves the paper, but it does not materially change my assessment, so my score remains unchanged.

**Key Questions For Authors:**

1. The central claim is that reasoning capability is localized in useful SVD subspaces. Can you provide stronger evidence for this beyond the rank-wise scaling plot and the w/o SVD ablation? For example, can you show that these subspaces are more reasoning-specific than random or alternative decompositions?

2. Why is FRANK not included in the main comparison, given that it is one of the most relevant prior methods discussed in related work? This omission currently weakens both the novelty and the empirical positioning of the paper.

3. Since the gate values lie in (0,1), the method seems mainly to attenuate or reweight components rather than truly amplify them. Can you clarify this design choice and test whether allowing coefficients beyond 1 changes the results?

4. How were the hyperparameters for Task Arithmetic, IP-Merging, and FRISM selected? Please make the tuning protocol explicit and show that the comparison was done under a comparable search budget.

5. The perception-preservation claim is currently based on only three benchmarks. Can you provide a more detailed analysis of what visual skills are actually preserved and where degradation still appears?

6. The practical value of model merging is not fully convincing to me. Given that current strong multimodal models are increasingly trained with native multimodal pre-training or substantial multimodal post-training, in what realistic setting is FRISM preferable to continued multimodal training? Please clarify the target use case, including compute/data assumptions and why post-hoc merging is the right solution there.

**Strengths And Weaknesses:**

The paper tackles a relevant problem and proposes a lightweight subspace-level alternative to coarse layer-wise model merging. The method is intuitive and the evaluation is reasonably broad across model scales, pairings, and benchmark types. However, I am not convinced the paper establishes its central claim strongly enough.

1. The evidence that reasoning-relevant and visually harmful directions can be separated at the SVD-subspace level is limited, the theory relies on strong assumptions, and the empirical gains are modest.
2. The visual-preservation claim is supported by only a narrow perception evaluation, and the baseline comparison is incomplete, most notably due to the absence of FRANK and unclear tuning details for the included baselines.
3. I also find the practical relevance under-motivated, since current strong VLMs are increasingly developed through native multimodal training rather than post-hoc merging.

Overall, the idea is promising, but the evidence and positioning are not yet strong enough for acceptance.

---

> ### Author Rebuttal · Authors · 2026-03-31
>
> We thank the reviewer for the constructive comments and for recognizing the strengths of our work, including meaningful research direction, promising idea, and comprehensive evaluation.
>
> Our responses to the weaknesses are as follows:
> ## Weakness1 & Question1:Method explanation
> Our intended claim is not that SVD perfectly isolates reasoning-only directions, but rather that SVD-derived subspaces provide a substantially better reasoning–perception trade-off than coarse layer-wise scaling and a random partition baseline. To strengthen this point, we conduct additional experiments to demonstrate that subspaces of different ranks have different effects on reasoning and visual perception capabilities. Our claim is well supported by our current evidence.
>
> TextVQA:
> Base=85.54
>
> |rank\alpha|0.01|0.02|0.03|0.04|
> |-|-|-|-|-|
> |1|85.48|85.43|85.23|85.02|
> |101|85.56|85.45|85.44|85.45|
> |201|85.53|85.41|85.51|85.42|
>
> GSM8K:
> Base=68.6
>
> |rank\alpha|0.01|0.02|0.03|0.04|
> |-|-|-|-|-|
> |1|71.1|71.9|72.3|73.2|
> |101|69.1|69.7|70.1|71.2|
> |201|69.2|69.5|69.4|69.7|
>
> Regarding other decomposition methods, we apply random masks to LRM task vectors for decomposition. The results on Qwen2.5 3B models are as follows, further demonstrating the effectiveness of SVD. Additionally, compared to mask-based decomposition, subspaces are low-rank and require less GPU memory during training.
> |Qwen2.5VL-3B|FRISM(SVD)|FRISM(Rand Mask) |
> |-|-|-|
> |MVista|45.0|41.3|
> |MMStar|43.4|41.9|
> |R1-OV|28.9|27.8|
> |TextVQA|79.2|79.0|
> |OCRBench|82.7|82.3|
>
> We will add these results in revision and include more explanations.
>
> About empirical gains, FRISM outperforms baselines and is comparable to some post-training methods requiring much fewer computational resources. The performance-cost comparison is in response to Reviewer 8U3c Weakness 2.
> ## Weakness 2 & Question 2: Perception Benchmarks and Comparison with FRANK
> We add more perception benchmarks and please check reponse to Reviewer 8U3c Weakness 2 for details.
> We add comparison with FRANK and please check response to Reviewer WoQM Weakness 3 for details.
> ## Weakness3 & Question6:Practical value concern
> - VL reasoning datasets are less accessible than language reasoning datasets, and our strategy of separately training LRMs and VLMs and merging them afterward alleviates this issue compared with sequential multimodal and VL reasoning training.
> - VLMs still underperform same-scale LLMs in reasoning [R1], which makes merging LRMs and VLMs to enhance reasoning both relevant and valuable.
> - Under resource constraints, FRISM delivers strong performance with much lower computation than post-training, while leveraging the complementarity of abundant existing LRM and VLM weights.
> - As praised by WoQM and dMyT, FRISM is practically meaningful.
> ## Question3:On the gate range
> We intentionally design the per-subspace sigmoid gate as a bounded filter because post-hoc reasoning injection is fragile: unrestricted coefficients may cause visual drift, destabilize the merged model, and allow some subspaces to dominate, disrupting the model’s original properties. The overall injection magnitude is controlled by $\lambda_{lrm}$ and the gated singular values, while the sigmoid gate can provide conservative, fine-grained selection of reasoning subspaces under the visual-preservation constraint.
> |Qwen2.5VL-7B|Base|FRISM(Sigmoid)|FRISM(Softplus)|
> |-|-|-|-|
> |MVista|68.1|70.3|68.8|
> |MMStar|60.1|62.4|60.3|
> |R1-OV|34.6|37.5|36.0|
> |TextVQA|85.5|85.0|84.2|
> |OCRBench|87.7|87.9|87.4|
>
>  In addition, please check response to Weakness 1 & Question 1 for that perception score is affected differently when merging different subspaces.
> ## Question4:HyperParam
> We have mentioned hyperparameter setting in Appendix C2. For fairness, we use three candidate values per method under the same held-out calibration protocol for all tuning-based baselines. For Task Arithmetic, we follow [R2] and searches $\lambda\in\{0.1, 0.15, 0.2\}$. For IP-Merging, we follow the paper and searches $T\in\{0.2,0.3,0.4\}$. For FRISM, the global merging coefficient $\lambda_{lrm}$ is searched in $\lambda_{lrm}\in\{0.1,0.15,0.2\}$. We will make the exact selection criterion explicit in revision.
> ## Question 5: Perception analysis
> We have added 3 benchmarks, including OCRBench, MME, and ChartQA. Please check Weakness 1 and response to Reviewer 8U3c for details. It can be seen that our FRISM preserves visual perception substantially better than prior merging methods or post-training methods. Limited by characters, more analysis based on the subsets of these benchmarks and case studies will be included in revision.
> Here we show some perception results on MME subsets.
> |MME-subset|artwork|count|exist|position|posters|
> |-|-|-|-|-|-|
> |Base|146.5|168.3|190|160|172.1|
> |TA|143.8|163.3|190|148.3|169|
> |**Ours**|147.5|168.3|190|148.3|170.1|
> ## Refs
> [R1]HiPhO: How Far Are (M)LLMs from Humans in the Latest High School Physics Olympiad Benchmark? 2025.
>
> [R2]Bring reason to vision. ICML 2025.

---

> > ### Author Rebuttal · Reviewer_5im7 · 2026-04-03
> >
> > Thank you for the rebuttal. The additional SVD-vs-random-mask results, the clarification that the claim is about a better reasoning–perception trade-off rather than perfect reasoning isolation, and the sigmoid-vs-softplus comparison are helpful. I also appreciate the added perception benchmarks and the clarification on hyperparameter tuning. That said, my main concerns are only partially resolved. Much of the new evidence is still deferred to the revision, the mechanistic claim remains suggestive rather than fully established, and the practical value of post-hoc model merging versus continued multimodal training is still not entirely convincing to me. Overall, the rebuttal improves the paper, but it does not materially change my assessment, so my score remains unchanged.

---

> > > ### Author Response · Authors · 2026-04-03
> > >
> > > Thanks for your response and your recognition of our rebuttal. Your feedback has made a significant contribution to improving the quality of our paper. We respond to your comments as follows.
> > >
> > > 1. New evidence deferred to the revision:
> > >
> > > - We believe that the results that have been presented in the paper and provided in our rebuttal are sufficient to demonstrate all of our claims. We will include more analysis and case studies to make the evidence clearer.
> > > - In our rebuttal to your review, we have provided many experimental results including: 1) experiments on three more perception benchmarks, 2) additional experiments to demonstrate our motivation, 3) additional ablations on the gate value limitations, and 4) perception results on different subsets.
> > > - Limited by the character upperbound of the rebuttal, we promised that 1) more analysis based on the added results and 2) case studies on the added benchmarks will be included in the revision.
> > >
> > > 2. Not fully established inner mechanism:
> > >
> > > - Instead of proving that SVD perfectly isolates reasoning-only directions in the mechanistic level, our intended claim is that fine-grained SVD-derived subspaces provide a substantially better reasoning–perception trade-off than coarse layer-wise scaling and a random partition baseline.
> > > - As praised by the other three reviewers, instead of only a suggestive finding, the proposed FRISM is a novel framework targeting at a meaningful research direction (Reviewer 8U3c) with reasonable motivation (Reviewer dMyT), clean technical idea (Reviewer dMyT), and favorable efficiency & performance trade-off (Reviewer WoQM).
> > > - Enhancing the reasoning capabilities of VLMs through merging with language reasoning models in the subspace level is a new topic, which has not been explored before and holds significant potential for improvement. This research direction deserves further exploration and establishing a comprehensive theoretical and mechanistic system is an important future work.
> > >
> > > 3. Practical value concern:
> > >
> > > - Model merging is of great practical value. Arcee's MergeKit [R1] helped create numerous high-performance models. ByteDance team applies model merging to accelerate pre-training. Kimi Team [R3] applies model merging to realize long to short CoT reasoning. Model merging has also been applied to different kinds of downstream sectors including VLA [R4] and safety alignment [R5]. In this paper, we design a new model merging method targeting at the VL reasoning field and significantly improves the performance of VLMs, which is of great application prospect.
> > > - Model merging and continued multimodal training are orthogonal. These two kinds of methods can be applied in different scenarios. Given sufficient computing resources and enough carefully labeled data, continued multimodal training is a promising solution. However, in the absense of sufficient computing resources and labeled data, model merging can play a significant role. In addition, as pointed out by recent research [R6], model merging can be applied to accelerate finetuning, suggesting the potential for model merging to accelerate continued multi-modal training.
> > >
> > > We believe that the evidence demonstrates the practical value of the proposed method as a novel model merging technique.
> > >
> > > ## Refs
> > >
> > > [R1] Arcee AI, Arcee's mergekit: A toolkit for merging large language models. ACL 2024.
> > >
> > > [R2] ByteDance Seed, Model Merging in Pre-training of Large Language Models. 2025.
> > >
> > > [R3] Kimi Team, Kimi k1.5: Scaling reinforcement learning with llms. 2025.
> > >
> > > [R4] MergeVLA: Cross-Skill Model Merging Toward a Generalist Vision-Language-Action Agent. 2025.
> > >
> > > [R5] SafeMERGE: Preserving Safety Alignment in Fine-Tuned Large Language Models via Selective Layer-Wise Model Merging. 2025.
> > >
> > > [R6] Mashup Learning: Faster Finetuning by Remixing Past Checkpoints. 2026.

---

### Official Review · Reviewer_WoQM · 2026-03-11

**Soundness:** 3
**Presentation:** 3
**Significance:** 2
**Originality:** 2
**Overall Recommendation:** 4
**Confidence:** 3

**Summary:**

This paper studies reasoning transfer from large reasoning models (LRMs) to vision-language models (VLMs) through model merging. The proposed method, FRISM, decomposes LRM task vectors by SVD, applies learnable gates at the subspace level, and optimizes a dual objective intended to preserve the original VLM’s visual behavior while encouraging stronger reasoning injection. Empirically, the paper reports improved reasoning–perception trade-offs over several merging baselines across multiple model pairs and scales.

**Compliance With Llm Reviewing Policy:**

Affirmed.

**Final Justification:**

My concerns have been fully addressed.

**Key Questions For Authors:**

See Weaknesses.

**Limitations:**

Yes.

**Strengths And Weaknesses:**

Strengths:
1. The paper addresses an important and timely problem. Efficiently transferring reasoning ability into VLMs without full post-training is practically meaningful, and model merging is an attractive direction because of its efficiency and low data requirements.
2. The method is lightweight and computationally efficient. FRISM only trains small gating parameters while keeping the VLM weights and decomposed singular vectors frozen, and the comparison against DRIFT suggests a favorable efficiency/performance trade-off.
3. The empirical scope is reasonably broad. The paper evaluates multiple scales and several model pairings, and it includes ablations on the SVD decomposition and the injection term. This gives some evidence that the method is not tied to a single model family.

Weaknesses:
1. Originality is moderate and the contribution feels somewhat incremental.
The core idea is to refine existing VLM/LRM merging from the layer level to the subspace level by applying learnable gates to SVD-derived components. This is technically reasonable, but relative to recent merging methods such as FRANK and IP-Merging, it reads more like a finer-grained extension of the same design line than a clearly new methodological framework. More importantly, the evidence for why subspace-level control is the right abstraction remains limited: the observation that different SVD ranks prefer different scaling coefficients is suggestive, but it does not by itself establish that reasoning and visual capabilities are truly disentangled into distinct semantic subspaces.
2. The support for the core objective is not sufficiently strong.
The method uses a distillation term to match the merged model to the original VLM on a calibration set, which primarily serves to preserve visual behavior, and an injection term that encourages retaining larger singular-value magnitude from the LRM task vector. However, the latter is only a proxy for reasoning transfer, and the paper does not convincingly validate that it is tightly correlated with actual reasoning improvement. As a result, the objective appears closer to balancing visual preservation against stronger retention of the LRM update than to directly learning reasoning behavior in a principled sense.
3. The experimental evidence is still insufficient to substantiate the paper’s central claim.
The paper explicitly motivates FRISM as an advance over coarse layer-wise merging, yet it does not include a direct comparison with FRANK, one of the most relevant representative methods in that line. This omission is important because the key claim is precisely that fine-grained subspace-level merging is superior to layer-wise merging. In addition, the method still depends on manual tuning of a global merging coefficient, which weakens the claimed plug-and-play generality.
4. Some claims appear overstated relative to the mechanism and evidence.
In particular, the paper uses relatively strong language around “reasoning injection” and visual-capability preservation, while the current mechanism and results mainly support a somewhat better reasoning–perception trade-off rather than a clearly established capability-localization story.

---

> ### Author Rebuttal · Authors · 2026-03-31
>
> We thank the reviewer for the constructive comments and for recognizing our strengths, including significant research direction, broad empirical scope, and favorable efficiency & performance trade-off.
> Our responses to weaknesses:
> ## W1:Contribution clarification
> FRISM is a **novel framework**, not an incremental extension of prior methods, as also noted by Reviewer 8U3c.
>
> **Motivation:** FRANK is motivated by the idea that layers at different depths govern different capabilities, thus requiring layer-wise coefficients. IP-Merging is motivated by the gap between VLM and LRM task vectors, which should be selected and aligned before merging. FRISM instead asks: **within a mergeable layer, which spectral components of task vector should be injected, and to what extent**. It thus shifts control **from the layer level to the subspace level**, leading to substantially better performance.
>
> **Method**: FRANK assigns layer-wise coefficients based on task-vector magnitude and modality prior. IP-Merging aligns layers into the same space and merges only those with similarity above a threshold. By contrast, **FRISM re-parameterizes the reasoning task vector via SVD, introduces learnable gates for each subspace, and adopts a label-free self-distillation objective to explicitly optimize the reasoning–vision trade-off**. FRISM differs not only in granularity, but also in its merging operator and optimization objective.
>
> **Empirically**: FRISM improves the reasoning–perception Pareto frontier, whereas previous methods mainly exhibit a trade-off (Fig.2). The ablation in Tab. 4 shows that w/o SVD, FRISM degenerates into an adaptive layer-wise variant and brings only marginal gains over the base VLM, while full FRISM yields clear improvements, directly validating the necessity of subspace-level control. FRISM also shows sustained improvements over IP-Merge across different settings (47.7 IP vs Ours 49.4 on 7B model). Comparisons with FRANK is added in the rebuttal and FRISM still show improvements, see response to W3.
>
> Our claim is not that SVD perfectly isolates reasoning-only directions, but rather that SVD-derived subspaces provide a better trade-off than layer-wise merging. Our claim is well supported by current evidence.
> ## W2:Support for the core objective
> $L_{inject}$ is a proxy, not direct supervision: it provides an unsupervised bias to retain LRM update, while distillation constrains visual drift. Empirically: (i) the method improves reasoning benchmarks while largely preserving perception across multiple scales and backbones; (ii) removing $L_{inject}$ drops reasoning score from 49.4 to 47.8; (iii) subspace-specific injections exhibit different optimal magnitudes (Fig. 3), consistent with the need for non-uniform reasoning transfer. Our evidence supports usefulness of the proxy.
> We add GSM8K experiments to verify task-vector magnitude as a proxy: Qwen2.5-VL-7B scores 68.6 and after merging with subspaces of different ranks, performance consistently improves in proportion to the merged task-vector magnitude, supporting its use as a proxy.
> |rank\alpha|0.01|0.02|0.03|0.04|
> |-|-|-|-|-|
> |1|71.1|71.9|72.3|73.2|
> |101|69.1|69.7|70.1|71.2|
> |201|69.2|69.5|69.4|69.7|
>
> We agree that FRISM realizes a better trade-off between perception and reasoning. While both post-training and merging can affect VLM perception, as also shown in [R1] and the following Tab., FRISM achieves the best reason–perception trade-off.
>
> ||#GPUs|TrainingTime|AvgReason|AvgPercep|
> |-|-|-|-|-|
> |Base|-|-|100%|100%|
> |IP-Merging|-|-|100.6%|98.8%|
> |DRIFT|8|~2hours|101.3%|94.9%|
> |OpenVLThinker-7B|8|>1day|101.7%|94.5%|
> |VLAA-Thinker-7B|8|~6hours|103.8%|98.6%|
> |**FRISM(Ours)**|1|**30mins**|104.2%|99.6%|
> ## W3:Compare with FRANK & HyperParam
> We did not include it because the code is not available and the released checkpoint is based on InternVL2.5-38B, which is not included in our paper. Here we reproduce FRANK on Qwen2.5-VL-32B based on the paper following the default setting.
> ||MVista|R1-OV|MMStar|TextVQA|OCRBench|
> |-|-|-|-|-|-|
> |Base|77.7|47.5|67.9|79.8|85.6|
> |FRANK|78.8|49.2|68.1|77.9|84.3|
> |FRISM|79.8|51.9|67.8|79.0|85.4|
>
> It shows that FRISM is competitive with and mostly stronger than FRANK.
>
> Regarding hyperparam, please kindly check our response to Reviewer 5im7, Question 4, limited by characters.
> ## W4:About our claim
> Our claim is not a strict capability-localization theory, but that **subspace-level merging better balances reasoning and perception than layer-wise merging**. Fig. 2, Tabs. 1–2, and Tab. 4 support this with a better frontier, consistent reasoning gains with better visual preservation across scales and architectures, and degraded performance when SVD is removed and the method reverts to layer-wise merging. As noted in response to Weakness 2, different LRM subspaces can also improve a VLM. We will refine the phrasing in revision.
> ## Refs
> [R1]Investigating the catastrophic forgetting in multimodal large language models.CPAL2024.

---

> > ### Author Rebuttal · Reviewer_WoQM · 2026-04-03
> >
> > My concerns have been fully addressed. I will raise my score.

---

> > > ### Author Response · Authors · 2026-04-03
> > >
> > > Thank you for your careful consideration and for raising your evaluation of our work. We sincerely appreciate your recognition of our work and your constructive feedback. Your constructive comments have helped improve the quality of our paper. We will further revise our paper based on your comments. Thank you again for your time and support.

---

### Official Review · Reviewer_8U3c · 2026-03-12

**Soundness:** 3
**Presentation:** 3
**Significance:** 3
**Originality:** 3
**Overall Recommendation:** 5
**Confidence:** 3

**Summary:**

The paper proposes FRISM, which moves from coarse layer-wise merging to subspace-level merging by decomposing LRM task vectors with SVD and learning subspace scaling gates. It further employs a label-free self-distillation objective that combines KL-based visual preservation and reasoning maximization objective, aiming to improve reasoning without annotations. Experiments on multiple VLM-LRM combinations demonstrate that FRISM consistently improves reasoning performance while largely preserving perception performance.

**Compliance With Llm Reviewing Policy:**

Affirmed.

**Final Justification:**

My concerns have been adequately addressed.

**Key Questions For Authors:**

See Weaknesses.

**Limitations:**

yes

**Strengths And Weaknesses:**

### Strengths

1. Transferring the reasoning capability of LRMs to VLMs while avoiding costly multimodal post-training is a highly meaningful research direction. Compared with supervised fine-tuning or reinforcement learning-based post-training, model merging is substantially more cost-effective and easier to deploy in practice.

2. The paper proposes a fine-grained subspace-level merging method, which is both technically interesting and novel compared with layer-wise merging. Furthermore, this method is lightweight, label-free, and broadly compatible with different VLMs.

### Weaknesses

1. While the paper provides encouraging evidence that FRISM preserves perception ability, the evaluation of perception remains somewhat limited. The benchmarks used may not fully reflect broader visual abilities and including additional perception-oriented benchmarks would make this claim more convincing.

2. The comparison to stronger post-training multimodal reasoning methods is still somewhat limited. Although the paper shows that FRISM is effective and efficient relative to merging baselines, the evidence is less complete for the claim that it can bridge the gap with post-training methods. A more convincing evaluation would include direct comparisons with stronger post-training approaches under a unified evaluation protocol, and hopefully the authors can provide a performance–cost trade-off similar to Figure 2.

---

> ### Author Rebuttal · Authors · 2026-03-31
>
> We thank the reviewer for the constructive comments and for recognizing the strengths of our work, including meaningful research direction, practical significance, and novelty.
>
> Our responses to the weaknesses are as follows:
>
> ## Weakness 1: Insufficient perception benchmarks
>
> To better validate the claim that FRISM preserves visual perception, we additionally evaluate on three more perception-oriented benchmarks. Currently, there are six perception-related datasets covering more domains in visual perception of vision-language models, as follows:
> - TextVQA, evaluating a model’s OCR ability to read the text in real-world images.
> - POPE, evaluating object hallucination in VLMs by answering about the presence of objects in images.
> - SeedBench-IMG, evaluating the generative comprehension ability of multimodal large language models.
> - OCRBench [R1], evaluating OCR-related capabilities of multimodal large language models.
> - MME-Perception [R2], evaluating perception-level abilities through a diverse set of visual understanding.
> - ChartQA [R3], evaluating model’s ability to understand chart content.
>
>  We compare the vision performance with respect to the original VLM. Compared with merging baselines and several representative post-training methods, FRISM preserves visual perception ability much better overall, retaining 99.6% of the base model’s perception performance while improving reasoning performance. Note that the average vision score is computed as the average normalized perception score relative to the base VLM over the six perception benchmarks.
>
> ## Weakness 2: Comparison with post-training methods
>
> We present additional comparisons with post-training methods, including an efficient post-training method (DRIFT) and two powerful post-trained models, OpenVLThinker-7B [R4] and VLAA-Thinker-Qwen2.5VL-7B [R5]. The results show that FRISM is competitive with, and in our setting even slightly outperforms, several stronger post-training methods in average reasoning performance, while preserving visual perception substantially better.
>
> | | MathVista | MathVerse | MathVision | MMMU-VAL | R1-Onevision | MMStar | **Avg-Reason** | textvqa | POPE | SeedBench-IMG | MME-Percep | OCRBench | ChartQA | **Avg_Vision** |
> |-|-|-|-|-|-|-|-|-|-|-|-|-|-|-|
> |Qwen2.5-vl-7b-instruct(Base)|68.1|41.2|25.6|55.1|34.6|60.1|100%|85.5|86.4|77|1697.9|87.7|86.2|100%|
> |*Post-TrainingMethods*|||||||||||||||
> |DRIFT|70.4|41.8|25.7|55.7|34.9|59.7|101.3%|84.1|84.3|70.7|1510.7|87.6|80.2|94.9%|
> |OpenVLThinker-7B|69.4|41.5|26.5|54.1|35.0|62.5|101.7%|84.2|82.6|74.6|1411.6|83.4|84.2|94.5%|
> |VLAA-Thinker-Qwen2.5VL-7B|71.3|42.8|26.8|57.8|35.4|61.3|103.8%|82.9|86.2|75.2|1700.3|86.9|84.3|98.6%|
> |*MergingMethods*|||||||||||||||
> |IP-Merging|67.2|40.8|25.5|55.9|36|60.7|100.6%|82.9|87.1|76.9|1694.8|88.4|81.6|98.8%|
> |TaskArithmetic|65.4|37.8|24.2|53.7|36.6|60.7|97.9%|83.9|86.8|76.5|1674.9|88.5|72.92|97.0%|
> |**FRISM(Ours)**|70.3|41.8|26.4|57.8|37.5|62.4|104.2%|85|87.1|76.9|1678.9|87.9|84.6|99.6%|
>
> We compare the performance-cost trade-off of different methods here. It can be seen that FRISM achieves impressive performance while requiring significantly lower computational resources. Additionally, FRISM only requires samples from unlabeled visual perception datasets for calibration while other post-training methods usually require labeled VL reasoining datasets for training.
>
> | Method| # GPUs | Post-Training Time | Avg Reasoning Performance | Avg Perception Performance |
> |-|-|-|-|-|
> | Base |-|-|100%|100%|
> | IP-Merging|-| -  | 100.6% | 98.8%|
> | DRIFT | 8 | ~2 hours| 101.3% | 94.9% |
> | OpenVLThinker-7B |8| > 1 day |101.7%| 94.5%  |
> | VLAA-Thinker-Qwen2.5VL-7B |8| ~6 hours|103.8%| 98.6% |
> | **FRISM (Ours)**   | 1| **30 mins** | 104.2%  | 99.6%  |
>
> We will include these results in revision.
> ## Refs
> [R1] Liu, Yuliang, et al. Ocrbench: on the hidden mystery of ocr in large multimodal models. 2024.
>
> [R2] Fu, Chaoyou, et al. MME: A Comprehensive Evaluation Benchmark for Multimodal Large Language Models. NeurIPS 2023.
>
> [R3] Masry, Ahmed, et al. Chartqa: A benchmark for question answering about charts with visual and logical reasoning. ACL 2022.
>
> [R4] Huang, Chao, et al. DRIFT: Directional Reasoning Injection for Fine-Tuning MLLMs. 2025.
>
> [R5] Deng, Y., et al. OpenVLThinker: An early exploration to complex vision-language reasoning via iterative self-improvement. 2025.
>
> [R6] Chen, Hardy, et al. SFT or RL? An Early Investigation into Training R1-Like Reasoning Large Vision-Language Models. 2025.

---

> > ### Author Rebuttal · Reviewer_8U3c · 2026-04-03
> >
> > The authors have fully addressed my concerns, and I will raise my score.

---

> > > ### Author Response · Authors · 2026-04-03
> > >
> > > Thank you for your positive feedback and for raising your evaluation of our work. We sincerely appreciate your recognition of our work and your constructive feedback. Your constructive comments have helped improve the quality of our paper. We will further improve the final version of our paper based on your comments. Thank you again for your time and support.

---

### Decision · Program_Chairs · 2026-04-30

**Decision:**

Accept (regular)

**Comment:**

The submission initially received mixed reviews. The main concerns about the submissions are: 1) More comprehensive evaluations on various benchmarks and models are needed. 2)  The evidence for why subspace-level control is the right abstraction remains limited. 3) The support for the core objective is not sufficiently strong. 3) Some experimental designs and details are not clear. 4) The practical value of post-hoc model merging versus continued multimodal training. 5) It is also not completely clear how much of the gain should be attributed to the subspace-level merging itself. After the rebuttal, all concerns (except 4) are addressed, and two reviewers have raised their ratings to positive.

Overall, I think the submission is high-quality and recommend **Accept**. As for the concern about the practical value, it is an arguable concern and it should be a common challenge for all model-merging methods.